# Graph Contrastive Learning under Heterophily via Graph Filters

**Wenhan Yang**[1]                    **Baharan Mirzasoleiman**[1]

[1]Computer Science Department, University of California Los Angeles (UCLA)

## Abstract

Graph contrastive learning (CL) methods learn node representations in a self-supervised manner by maximizing the similarity between the augmented node representations obtained via a GNN-based encoder. However, CL methods perform poorly on graphs with heterophily, where connected nodes tend to belong to different classes. In this work, we address this problem by proposing an effective graph CL method, namely HLCL, for learning graph representations under heterophily. HLCL first identifies a homophilic and a heterophilic subgraph based on the cosine similarity of node features. It then uses a low-pass and a high-pass graph filter to aggregate representations of nodes connected in the homophilic subgraph and differentiate representations of nodes in the heterophilic subgraph. The final node representations are learned by contrasting both the augmented high-pass filtered views and the augmented low-pass filtered node views. Our extensive experiments show that HLCL outperforms state-of-the-art graph CL methods on benchmark datasets with heterophily, as well as large-scale real-world graphs, by up to 7%, and outperforms graph supervised learning methods on datasets with heterophily by up to 10%.

## 1 INTRODUCTION

Graph neural networks (GNNs) are powerful tools for learning graph-structured data in various domains [Kipf and Welling, 2016, Veličković et al., 2017]. GNNs use the graph's adjacency matrix to aggregate node information from their neighbors, effectively acting as a low-pass filter that smooths graph signals [Nt and Maehara, 2019]. They have shown remarkable success in supervised and semi-supervised learning, where task-specific labels are available. However, obtaining high-quality labels can be costly in many domains, spurring interest in self-supervised learning on graphs to learn representations without supervision [Velickovic et al., 2019, Peng et al., 2020, Qiu et al., 2020, Hassani and Khasahmadi, 2020, Zhu et al., 2020b].

Among these self-supervised methods, Contrastive Learning (CL) has demonstrated remarkable success [Velickovic et al., 2019, Peng et al., 2020, Qiu et al., 2020, Hassani and Khasahmadi, 2020, Zhu et al., 2020b]. Graph CL methods first augment the input graph, either by altering node features or the graph topology. Then, they learn representations by contrasting the augmented graph views encoded with a GNN-based encoder. Existing graph CL methods perform well under homophily, where neighboring nodes often share the same label. However, they perform poorly on heterophilic graphs, where connected nodes tend to belong to different classes [Zhu et al., 2020b]. Indeed, for learning rich representations in graphs with heterophily, contrasting augmented views of every node is not enough, but it is crucial to *differentiate* representation of node with different labels [Bo et al., 2021, Luan et al., 2020]. However, *without label* information, it is not clear how this can be achieved.

In this work, we propose an effective graph CL method, namely HLCL, for learning node representations under heterophily. HLCL first uses nodes' feature similarity to identify a homophilic and heterophilic subgraph in the original graph. Then, for each subgraph, it generates two augmented graph views, and applies a high-pass filter to the heterophilic subgraphs and a low-pass filter to the homophilic subgraphs. The final representations are learned by contrasting the augmented high-pass filtered views and contrasting the augmented low-pass filtered views of each node, using the *same GNN encoder*, as illustrated in Fig. 1. In doing so, HLCL achieves state-of-the-art performance under heterophily, surpassing graph *supervised learning* methods and yielding comparable performance to state-of-the-art graph CL methods under homophily. In addition, we prove that the learned representations by HLCL encode both low-frequency and

*Accepted for the 40th Conference on Uncertainty in Artificial Intelligence* (UAI 2024).

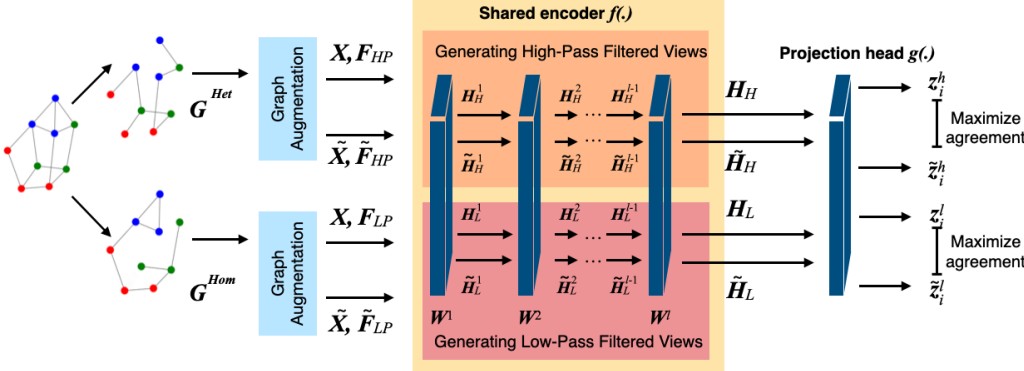

Figure 1: HLCL identifies a homophilic and a heterophilic subgraph $\mathcal{G}^{hom}, \mathcal{G}^{het}$, and generates two augmentations for each subgraph. Then, it applies low-pass filters $\boldsymbol{F}_{LP}, \tilde{\boldsymbol{F}}_{LP}$ to the augmented homophilic subgraphs and high-pass filters $\boldsymbol{F}_{HP}, \tilde{\boldsymbol{F}}_{HP}$ to the augmented heterophilic subgraphs, to generate low-pass $\boldsymbol{H}_L, \tilde{\boldsymbol{H}}_L$ and high-pass $\boldsymbol{H}_H, \tilde{\boldsymbol{H}}_H$ filtered views, using the same encoder $\boldsymbol{W}$. HLCL learns the final representations by contrasting the projected low-pass filtered augmented views $\boldsymbol{z}^L, \tilde{\boldsymbol{z}}^L$ and the high-pass filtered augmented views $\boldsymbol{z}^h, \tilde{\boldsymbol{z}}^h$ of every node.

high-frequency information.

Our extensive experiments show that on seven benchmark datasets, HLCL outperforms existing graph CL methods by up to 7% and graph supervised learning methods by up to 10% under heterophily, while maintaining comparable performance under homophily. Additionally, HLCL scales well to large graphs like Penn94, outperforming other graph CL methods by up to 5%.

In summary, our contributions are as follows:

- *Graph CL with graph filters.* HLCL is the first graph CL method that utilizes graph filters, and combines high-pass and low-pass filtered representations using contrastive losses. This approach enables learning rich representations under heterophily.
- *Careful aggregation.* HLCL identifies a homophilic and a heterophilic subgraph based on node features or representations, for effective information aggregation.
- *Theoretical analysis.* By analyzing HLCL, we theoretically prove that HLCL learns the invariance information from both homophilic and heterophilic subgraphs.
- *Extensive experiments.* Empirically, we confirm that HLCL achieves state-of-the-art under heterophily and a competitive performance under homophily.

## 2 RELATED WORK

**Graph self-supervised learning.** Graph self-supervised learning (SSL) methods have become a powerful tool for learning representations without any labels. Graph contrastive learning (CL) is among the most successful graph SSL methods. Numerous methods have been proposed in the field: [Velickovic et al., 2019, Peng et al., 2020, Hassani and Khasahmadi, 2020, Zhu et al., 2021c] focus on contrasting the global augmented representation with the local augmented representation, while [Zhu et al., 2020c,

You et al., 2020, Qiu et al., 2020, Liu et al., 2022] contrast same-scale representation, global or local, in two augmented views. Due to the complexity of collecting negative samples in graph data, negative-sample free contrastive objectives have also been studied [Thakoor et al., 2021, Bielak et al., 2021]. However, such work focus on encoding the homophilic graphs and perform poorly under heterophily. Recently, a stream of SSL methods have been proposed to learn the node representations of the heterophilic graphs without labels. HGRL [Chen et al., 2022] improves the node representations on heterophilic graphs by rewiring non-local neighbors based on feature information before training. SP-GCL [Wang et al., 2022] considers nodes from the $T$-hop neighborhood of a node with high feature similarities as positive pairs, without using any explicit augmentations. DSSL [Xiao et al., 2022] separates the heterogeneous patterns in local neighborhood distributions to capture both homophilic and heterophilic information globally. GREET [Liu et al., 2023] discriminates homophilic edges from heterophilic edges using random walk based graph diffusion and contrasts the projected representations of the two graph views directly via a dual-channel contrastive loss. MUSE [Yuan et al., 2023] creates two views to capture information from the node itself and its neighborhood, and fuses these views to enhance node representations. NeCo [He et al., 2023] proposes a new pretext task, group discrimination, which divides the nodes into $k$ groups and keeps the consistent representation of nodes within a group.

**Graph (semi-)supervised learning under heterophily.** In the supervised setting, recent methods propose to use other types of aggregation that better fit graphs with heterophily. Zhu et al. [2021a] analyzed and designed a uniform framework for GNNs' propagations and proposed GNN-LF and GNN-HF that preserve information of different frequency separately by using different filtering kernels with learnable weights. FAGCN [Bo et al., 2021] and FBGNN [Luan

et al., 2020] train two *separate* encoders to capture the high-pass and low-pass graph signals separately. Then they rely on labels to learn relatively complex mechanisms to combine the outputs of the encoders. However, learning how to combine the encoder outputs is highly sensitive to having high-quality labels. This makes such methods highly impractical for self-supervised contrastive learning, where the label information is not available. Unlike the above supervised methods, we apply the high-pass and low-pass filters to different subgraphs, contrasting the resulting high-pass filtered node views and low-pass filtered node views in a self-supervised manner, without any label. This is in contrast to learning the best combination of filtered signals of different encoders based on labels. A more comprehensive overview of related work are provided in Appendix A.6.

# 3 PRELIMINARIES

**Notations.** We denote by $\mathcal{G} = (\mathcal{V}, \mathcal{E})$ an undirected graph, where $\mathcal{V} = \{v_1, v_2, \ldots, v_N\}$ represents the node set, and $\mathcal{E} \subseteq \mathcal{V} \times \mathcal{V}$ represents the edge set. We denote by $\boldsymbol{A} \in \{0, 1\}^{N \times N}$ the symmetric adjacency matrix of graph $\mathcal{G}$, where $\boldsymbol{A}_{ij} = 1$ if and only if $(v_i, v_j) \in \mathcal{E}$, and $\boldsymbol{A}_{ij} = 0$ otherwise. We denote the feature matrix by $\boldsymbol{X}$, where $\boldsymbol{X}_{i.} \in \mathbb{R}^m$ is the feature vector of the $i^{th}$ node, and $\boldsymbol{x} \in \mathbb{R}^N$ is a column of the matrix and represents a graph signal. $\boldsymbol{D}$ is the degree matrix of the graph, with $\boldsymbol{D}_{ii} = \sum_j \boldsymbol{A}_{ij}$, and $\mathcal{N}_i = \{j : \boldsymbol{A}_{ij} = 1\}$ is the neighborhood of node $i$. $\boldsymbol{L}$ is the Laplacian matrix of the graph, defined as $\boldsymbol{L} = \boldsymbol{D} - \boldsymbol{A}$. The normalized Laplacian matrix is denoted by $\boldsymbol{L}_{sym} = \boldsymbol{D}^{-\frac{1}{2}} \boldsymbol{L} \boldsymbol{D}^{-\frac{1}{2}}$, and the normalized adjacency matrix is defined as $\boldsymbol{A}_{sym} = \boldsymbol{D}^{-\frac{1}{2}} \boldsymbol{A} \boldsymbol{D}^{-\frac{1}{2}}$. Here, we use the renormalized version of the adjacency matrix $\hat{\boldsymbol{A}}_{sym} = \bar{\boldsymbol{D}}^{-\frac{1}{2}} \bar{\boldsymbol{A}} \bar{\boldsymbol{D}}^{-\frac{1}{2}}$ as introduced in [Kipf and Welling, 2016], where $\bar{\boldsymbol{A}} = \boldsymbol{A} + \boldsymbol{I}$, $\bar{\boldsymbol{D}} = \boldsymbol{D} + \boldsymbol{I}$. Similarly, the renormalized Laplacian matrix is defined as $\hat{\boldsymbol{L}}_{sym} = \boldsymbol{I} - \hat{\boldsymbol{A}}_{sym}$. $\hat{\boldsymbol{L}}_{sym}$ is a real symmetric matrix, with orthonormal eigenvectors $\{\boldsymbol{u}_i\}_{l=1}^n \in \mathbb{R}^n$, and corresponding eigenvalues $\lambda_i \in [0, 2)$ [Chung, 1997]. For $\hat{\boldsymbol{A}}_{sym}$ we have $\lambda_i(\hat{\boldsymbol{A}}_{sym}) \in (-1, 1]$.

## 3.1 GRAPH CL UNDER HOMOPHILY

State-of-the-art graph CL methods explicitly augment the input graph using feature or topology augmentations, encode the augmented graphs using a GNN-based encoder, and contrast the encoded node representations [Zhu et al., 2020c, 2021b, Velickovic et al., 2019, Thakoor et al., 2021, Qiu et al., 2020], as we will discuss in more detail next.

**Graph Augmentation.** First, the input graph is explicitly augmented, by altering its topology or node features. Topology augmentation methods remove or add nodes or edges, and feature augmentation methods alter the node features by masking particular columns, dropping features at random, or randomly shuffling the node features [Zhu et al., 2020c,

2021b, Velickovic et al., 2019, Thakoor et al., 2021].

**GNN Encoder.** The augmented graphs are then passed through a GNN-based encoder to obtain the augmented node views. The GNN encoder produces node representations by aggregating the node features in a neighborhood as follows:

$$\boldsymbol{H}^l = \sigma(\tilde{\boldsymbol{A}}_{sym} \boldsymbol{H}^{l-1} \boldsymbol{W}^{l-1}), \quad \boldsymbol{H}^0 = \boldsymbol{X}, \qquad (1)$$

where $\boldsymbol{H}_L^l$ is the node representations at layer $l$ of the encoder, $\boldsymbol{W}^l \in \mathbb{R}^{d_l \times d_{l-1}}$ is the weight matrix in layer $l$ of the encoder, and $\sigma$ is the activation function. Crucially, the Adjacency matrix $\tilde{\boldsymbol{A}}_{sym}$ is a low-pass filter that aggregates every node's features with the features of nodes in its immediate neighborhood. For a multi-layer graph encoder, it iteratively aggregates features in a multi-hop neighborhood of every node to learn its representation. Hence, it smooths out the node representations and produces similar representations for the nodes within the same multi-hop neighborhood.

**Contrastive Loss.** Finally, the contrastive loss distinguishes the representations of the same node in two different augmented views, from other node representations. For example the commonly used InfoNCE loss [Oord et al., 2018] is:

$$-\log \frac{e^{\text{sim}_\tau(\boldsymbol{u}^i, \boldsymbol{v}^i)}}{e^{\text{sim}_\tau(\boldsymbol{u}^i, \boldsymbol{v}^i)} + \sum_{k \neq i} e^{\text{sim}_\tau(\boldsymbol{u}^i, \boldsymbol{v}^k)}}, \qquad (2)$$

where $\boldsymbol{u}_i, \boldsymbol{v}_i$ are representations of two different augmented views of node $i$, $\text{sim}(\boldsymbol{u}^i, \boldsymbol{v}^k)$ is the cosine similarity between $\boldsymbol{u}^i$ and $\boldsymbol{v}^k$, and $\tau$ is a temperature parameter.

## 3.2 HIGH-PASS AND LOW-PASS GRAPH FILTERS

The adjacency and Laplacian matrices can be leveraged to filter the smooth and non-smooth graph components, and capture similarity and dissimilarity of node features to their neighborhoods. Specifically, multiplication of Laplacian with a graph signal $\hat{\boldsymbol{L}}_{sym} \boldsymbol{x} = \sum_i \lambda_i \boldsymbol{u}_i \boldsymbol{u}_i^T \boldsymbol{x}$, acts as a filtering operation over $\boldsymbol{x}$, adjusting the scale of the components of $\boldsymbol{x}$ in the frequency domain. The entries of every eigenvector, $\boldsymbol{u}_i$ align with a cluster of connected nodes in the graph. For the Laplacian matrix, a smaller eigenvalue $\lambda_i$ corresponds to a lower frequency (smoother) eigenvectors $\boldsymbol{u}_i$, and a larger cluster of connected nodes. On the other hand, a larger $\lambda_i$ corresponds to a high frequency (non-smooth) eigenvectors $\boldsymbol{u}_i$, which identify smaller clusters of closely connected nodes in the graph. A Laplacian filter magnifies the high frequency signals that align well with basis functions corresponding to large eigenvalues $\lambda_i \in (1, 2)$ and suppresses the low frequency signal that aligns with basis functions corresponding to small eigenvalues $\lambda_i \in [0, 1]$. That means, for small clusters of nodes that have a large alignment with $\boldsymbol{u}_i$ corresponding to $\lambda_i > 1$, the projection $\lambda_i \boldsymbol{u}_i \boldsymbol{u}_i^T \boldsymbol{x}$ amplifies $\boldsymbol{x}$ within the cluster and consequently magnifies the difference in $\boldsymbol{x}$ among the nodes within that cluster. On the other hand, for the larger clusters

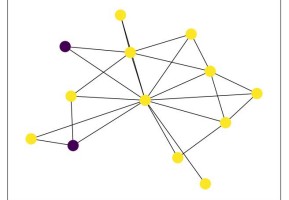 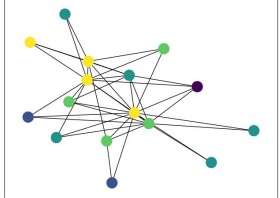

(a) Homophily neighborhood     (b) Heterophily neighborhood

Figure 2: Chameleon ($\beta$=0.23). Heterophilic graphs contain neighborhoods with homogeneous & heterogeneous labels.

that align well with $\boldsymbol{u}_i$ corresponding to $\lambda_i < 1$, the projection $\lambda_i \boldsymbol{u}_i \boldsymbol{u}_i^T \boldsymbol{x}$ suppresses $\boldsymbol{x}$ within the cluster and reduces the differences in $\boldsymbol{x}$ among the nodes within that cluster. Hence the Laplacian matrices can be generally regarded as high-pass filters [Ekambaram, 2014], that enlarge the differences in node features over small clusters, and smooths out the differences over larger clusters in the graph. In contrast, affinity matrices, such as the normalized adjacency matrix, can be treated as low-pass filters [Nt and Maehara, 2019], which suppress and filter out non-smooth components of the signals. This is because all of the eigenvalues of the affinity matrices are smaller than 1, i.e., $\lambda_i \in (-1, 1]$.

On the node level, left multiplying $\hat{\boldsymbol{L}}_{sym}$ and $\hat{\boldsymbol{A}}_{sym}$ filters with $\boldsymbol{x}$ can be understood as diversification and aggregation operations, respectively [Luan et al., 2020]. In particular, a typical GNN filters smooth graph frequencies by aggregating the node representations with those of their neighbors, using the adjacency matrix, i.e.,

$$(\hat{\boldsymbol{A}}_{sym}\boldsymbol{x})_i = \sum_{j \in \mathcal{N}_i} \frac{1}{\overline{\boldsymbol{D}}_{ii}} \boldsymbol{x}_j. \tag{3}$$

Hence, it results in similar representations for the nodes in a neighborhood. In contrast, the high-pass filter only preserves the high-pass frequencies, using the Laplacian matrix, i.e.

$$(\hat{\boldsymbol{L}}_{sym}\boldsymbol{x})_i = \sum_{j \in \mathcal{N}_i} \frac{1}{\overline{\boldsymbol{D}}_{ii}} (\boldsymbol{x}_i - \boldsymbol{x}_j). \tag{4}$$

In doing so, it magnifies the dissimilarities between the nodes and make the representations of nodes in a neighborhood distinguishable.

**Homophily Ratio** Homophily ratio quantifies how likely nodes with same labels are connected in the graph. Formally, homophily ratio, $\beta$, is defined as follows [Pei et al., 2020]:

$$\beta = \frac{1}{|V|} \sum_{v \in V} \frac{\text{No. of similar neighbors of } v}{\text{No. of neighbors of } v}. \tag{5}$$

## 4 GRAPH CL UNDER HETEROPHILY

In this section, we first discuss the challenges of having a universal method for graph CL under heterophily and

homophily. Then, we present our approach to overcome these challenges and learn high-quality representations.

**Challenges.** Under heterophily, where nodes in a neighborhood may have different labels, aggregating the node representations in a neighborhood fades out the dissimilarity between representations of node in different classes, and contrasting those augmented representations further makes them indistinguishable. Labels can help guide an appropriate aggregation in the neighborhood. However, without labels, it is not clear how the neighborhood information should be aggregated. Additionally, even if one can identify homophilic edges, the number of such edges may be too small to learn high quality representations via GNNs, under heterophily. To achieve rich representations in such graphs, it is crucial to not only aggregate representations of neighbors with the same label, but also push away representations of neighbors with different labels. This allows learning richer node representations based on both similarities and dissimilarities of the nodes in different neighborhoods.

Next, we present our method, HLCL, that can learn high-quality representations under heterophily.

### 4.1 HIGH-PASS & LOW-PASS GRAPH CL (HLCL)

As discussed, under heterophily, leveraging node feature similarities is not enough for learning high-quality representations. It is crucial to capture the *dissimilarities* between the neighboring nodes to separate different classes. A high-pass filter like the Laplacian matrix (*c.f.* Sec. 3.2) filters the non-smooth graph component and captures the dissimilarity of the node features in a neighborhood. However, without labels, we cannot know whether the graph is homophilic or heterophilic, and naively using a high-pass filter instead of a low-pass filter significantly harms the performance under homophily. Moreover, most heterophilic graphs also consist of several neighborhoods with homogeneous labels, as illustrated in Fig.2. Hence, simply applying a high-pass filter to an unlabeled graph may result in poor performance.

**Idea.** To learn rich node representations for both graph types, our main idea is to first identify a homophilic subgraph and a heterophilic subgraph in the original graph. Then, we augment each subgraph, and apply a low-pass filter to the augmented homophilic subgraphs and a high-pass filter to the augmented heterophilic subgraphs to obtain two high-pass and two low-pass filtered views for every node, using the *same encoder*. The final representations are learned by contrasting the two high-pass filtered views and the two low-pass filtered views of every node.

Next, we introduce our method, HLCL, which works based on the above idea.

**Separating Subgraphs.** HLCL first identifies two subgraphs in the original graph: a homophilic subgraph with

edges connecting nodes with homogeneous labels, and a heterophilic subgraph with edges connecting nodes with heterogeneous labels.

Formally, given a graph $\mathcal{G} = (\mathcal{V}, \mathcal{E})$, the heterophilic subgraph $\mathcal{G}^{het} = (\mathcal{V}, \mathcal{E}^{het})$ and the homophilic subgraph $\mathcal{G}^{hom} = (\mathcal{V}, \mathcal{E}^{hom})$ each contain all the nodes $\mathcal{V}$, and a subset of the edges of the original graph, i.e., $\mathcal{E}^{het}, \mathcal{E}^{hom} \subseteq \mathcal{E}$. We denote by $\boldsymbol{A}^{het}, \boldsymbol{A}^{hom} \in \{0,1\}^{N \times N}$ the symmetric adjacency matrix of subgraphs $\mathcal{G}^{het}, \mathcal{G}^{het}$, respectively. Note that the feature matrix $\boldsymbol{X}$ for $\mathcal{G}$ is the same as $\mathcal{G}^{het}$ and $\mathcal{G}^{hom}$. However, the neighborhood for a given node $i$ can be different in the two subgraphs. We define $\mathcal{N}_i^{het} = \{j : \boldsymbol{A}_{ij}^{het} = 1\}$ and $\mathcal{N}_i^{hom} = \{j : \boldsymbol{A}_{ij}^{hom} = 1\}$ as the neighborhood of node $i$ in $\mathcal{G}^{het}, \mathcal{G}^{hom}$, respectively. Without any label supervision, we rely on the important observation that for graphs with different homophily ratios, the original features can approximately indicate the label information [Jin et al., 2021, Chen et al., 2022, Wang et al., 2020, Zhu et al., 2020b]. Based on this observation, we calculate pairwise feature similarities $s_{ij} = \langle \boldsymbol{x}_{i.}, \boldsymbol{x}_{j.} \rangle$ for all $i, j \in [n] = |\mathcal{V}|$, where $\langle ., . \rangle$ is the cosine similarity. Then, we first form the homophilic subgraph by selecting $k_1$ fraction of edges in neighborhood of every node $i$ with largest cosine similarities. Formally, $\mathcal{E}^{hom} = \big\{(i,j)|i \in [n], j \in \arg\max_{P \subseteq \mathcal{N}_i, |S|=\lceil k_1 \cdot |\mathcal{N}_i|\rceil} \sum_{p \in P} \{s_{i,p}\}\big\}$. Next, we form the heterophilic subgraph using $k_2$ fraction of the edges in neighborhood of every node with lowest cosine similarities, i.e., $\mathcal{E}^{het} = \big\{(i,j)|i \in [n], j \in \arg\min_{P \subseteq \mathcal{N}_i, |S|=\lceil k_2 \cdot |\mathcal{N}_i|\rceil} \sum_{p \in P} \{s_{i,p}\}\big\}$.

The initial subgraphs are constructed via original node features. However, the subgraphs are updated every $T$ epochs with the learned node representations during training. Note that, in contrast to prior work [Chen et al., 2022], we do not introduce new edges based on feature similarities throughout the training, which could change the semantic information of the graph [He et al., 2023].

We note that all the nodes may not be connected in both subgraphs. However, as long as one subgraph is mostly connected, the information can be aggregated effectively and a satisfactory performance is obtained. For example, under homophily the heterophilic subgraph is small, but the homophilic subgraph contains almost all the nodes in the largest connected component of the original graph. Similarly, under extreme heterophily almost all the nodes are in the heterophilic subgraph and the homophilic subgraph is small and minimally affects the performance. As HLCL contrasts augmented views of the homophilic subgraph and heterophilic separately (it does not contrast the subgraphs with each other), only one subgraph needs to be mostly connected to achieve satisfactory performance. In our ablation studies in Sec. 6.5, we confirm that in real-world graphs at least one of the subgraphs are mostly connected.

**Augmenting the Subgraphs.** Next, HLCL generates two augmented views for each subgraph via random graph perturbations. We denote the two augmented graph views as $\mathcal{G}$ and $\tilde{\mathcal{G}}$. For the homophilic subgraph, we follow [Zhu et al., 2020c] and apply edge removal and feature masks as our graph augmentations. For the heterophilic subgraph, we apply node dropping and feature masks as our augmentations. We study the effects of different augmentation techniques on the heterophilic subgraph in Sec. 6.1.

**Producing the Filtered Representations.** Subsequently, HLCL applies a high-pass filter to the two augmented views of the heterophilic subgraph, and a low-pass filter to the two augmented views of the homophilic subgraph, using the *same encoder*. The shared encoder is crucial to ensure a good performance under both homophily and heterophily.

Specifically, to generate the low-pass and high-pass filtered node views, HLCL leverages the renormalized adjacency matrices of the augmented heterophilic subgraph and the renormalized Laplacian matrices of the augmented heterophilic subgraph. Formally, $\boldsymbol{F}_{LP} = \hat{\boldsymbol{A}}_{sym}^{hom}$, and $\boldsymbol{F}_{HP} = \hat{\boldsymbol{L}}_{sym}^{het} = \boldsymbol{I} - \hat{\boldsymbol{A}}_{sym}^{hom}$, are the low-pass and high-pass filters corresponding to the first augmented view of the homophilic and heterophilic subgraphs, and $\tilde{\boldsymbol{F}}_{LP}, \tilde{\boldsymbol{F}}_{HP}$ are the low-pass and high-pass filters corresponding to the second augmented view of the homophilic and heterophilic subgraphs. Effectively, $\boldsymbol{F}_{LP}, \tilde{\boldsymbol{F}}_{LP}$ are the aggregation operations in Eq. (3) and $\boldsymbol{F}_{HP}, \tilde{\boldsymbol{F}}_{HP}$ are diversification operations in Eq. (4). Then, the two low-pass filtered views of the homophilics subgraph are obtained as follows:

$$\boldsymbol{H}_L^l = \sigma(\boldsymbol{F}_{LP} \boldsymbol{H}_H^{l-1} \boldsymbol{W}^{l-1}), \qquad (6)$$

$$\tilde{\boldsymbol{H}}_L^l = \sigma(\tilde{\boldsymbol{F}}_{LP} \boldsymbol{H}_H^{l-1} \boldsymbol{W}^{l-1}), \qquad (7)$$

and the two high-pass filtered views of the heterophilic subgraph are obtained as follows:

$$\boldsymbol{H}_H^l = \sigma(\boldsymbol{F}_{HP} \boldsymbol{H}_H^{l-1} \boldsymbol{W}^{l-1}), \qquad (8)$$

$$\tilde{\boldsymbol{H}}_H^l = \sigma(\tilde{\boldsymbol{F}}_{HP} \boldsymbol{H}_L^{l-1} \boldsymbol{W}^{l-1}). \qquad (9)$$

$\boldsymbol{H}_L^l, \tilde{\boldsymbol{H}}_L^l$ are the low-pass filtered augmented views at layer $l$ of the encoder, $\boldsymbol{H}_H^l, \tilde{\boldsymbol{H}}_H^l$ are the high-pass filtered augmented views at layer $l$ of the encoder, $\boldsymbol{W}^l \in \mathbb{R}^{d_l \times d_{l-1}}$ is the weight matrix in layer $l$ of the encoder, $\sigma$ is the activation function, and we have $\boldsymbol{H}_L^0 = \boldsymbol{H}_H^0 = \boldsymbol{X}$, and $\tilde{\boldsymbol{H}}_L^0 = \tilde{\boldsymbol{H}}_H^0 = \tilde{\boldsymbol{X}}$ where $\boldsymbol{X}, \tilde{\boldsymbol{X}}$ are augmented feature matrices.

$\boldsymbol{F}_{HP}, \tilde{\boldsymbol{F}}_{HP}$ filter out the low-frequency signals and preserve the high-frequency signals. In doing so, they capture the difference in feature of each node and its neighbors. Using a high-pass encoder within a multi-layer encoder iteratively captures the difference between features of the nodes in a multi-hop neighborhood of a node in the heterophilic subgraph. Hence, it makes the representations of nodes that have different features from their neighbors distinct in their multi-hop neighborhood. On the other hand,

$\boldsymbol{F}_{LP}, \tilde{\boldsymbol{F}}_{LP}$, only preserve the low-frequency signals by aggregating every node's features with those of its immediate neighborhood. Using the low-pass filter within a multi-layer graph encoder iteratively aggregates features in a multi-hop neighborhood of every node in the homophilic subgraph to learn its representation. Hence, they smooth out the node representations and produces similar representations for the nodes within the same multi-hop neighborhood.

Note that, we use the Laplacian and adjacency matrices of the augmented subgraphs instead of those of the original graphs, as they indicate how the information in different neighborhoods should be aggregated by the GCN encoder. Indeed, it is important to use the corresponding matrices in the subgraphs. In doing so, we pull together representations of nodes within label-homogeneous neighborhoods by applying low-pass filters to homophilic subgraphs, and push away representations of nodes within label-heterogeneous neighborhoods by applying high-pass filter to heterophilic subgraphs. If both filters are applied to the original graph, representations of the nodes within each neighborhood will be pulled together and pushed apart at the same time.

Using both high-pass and low-pass filters provide complementary information and allow learning both smooth and non-smooth components of the graphs simultaneously, which is particularly useful for graphs under heterophily. We note that other types of high-pass and low-pass filters can be used in a similar way in our framework.

**Contrasting the Filtered Representations.** Finally, by contrastive the augmented views of each subgraph, HLCL learns high-quality representations. The augmented views $\boldsymbol{H}, \tilde{\boldsymbol{H}}$ are first projected via a 2-layer non-linear MLP, named projection head, to another latent space $\boldsymbol{z}, \tilde{\boldsymbol{z}}$ where the contrastive losses are calculated, as advocated by [Chen et al., 2020, Chen and He, 2021, Zhu et al., 2020c, 2021b].

Then, for each subgraph, we first consider every node $i$ in the first augmented subgraph view as the anchor, and contrast it with all the nodes in the second augmented subgraph view. This yields the following contrastive losses for the homophilic and heterophilic subgraphs, respectively:

$$l(\boldsymbol{z}_l^i, \tilde{\boldsymbol{z}}_l^i) = \log \frac{e^{\mathrm{sim}(\boldsymbol{z}_l^i, \tilde{\boldsymbol{z}}_l^i)/\tau}}{e^{\mathrm{sim}(\boldsymbol{z}_l^i, \tilde{\boldsymbol{z}}_l^i)/\tau} + \sum_{\substack{k \in [N], \\ k \neq i}} e^{\mathrm{sim}(\boldsymbol{z}_l^i, \tilde{\boldsymbol{z}}_l^k)/\tau}} \quad (10)$$

$$l(\boldsymbol{z}_h^i, \tilde{\boldsymbol{z}}_h^i) = \log \frac{e^{\mathrm{sim}(\boldsymbol{z}_h^i, \tilde{\boldsymbol{z}}_h^i)/\tau}}{e^{\mathrm{sim}(\boldsymbol{z}_h^i, \tilde{\boldsymbol{z}}_h^i)/\tau} + \sum_{\substack{k \in [N], \\ k \neq i}} e^{\mathrm{sim}(\boldsymbol{z}_h^i, \tilde{\boldsymbol{z}}_h^k)/\tau}}, (11)$$

where sim is the cosine similarity between the projected representations, and $\tau$ is a temperature parameter. The second term in the denominator represent the inter-view negative pairs, which are between the anchored view of node $i$ and the views of all other nodes from the other view.

Similarly, for each subgraph we also consider the second augmented view of node $i$ as the anchor and contrast it with

---

**Algorithm 1** High-pass and Low-pass Graph CL (HLCL)

1: Infer subgraph $\mathcal{G}^{hom}$ by selecting the top $\lceil k_1 \times |\mathcal{N}_i| \rceil$ edges with highest cosine similarity for every node $i$.
2: Infer subgraph $\mathcal{G}^{het}$ by selecting the top $\lceil k_2 \times |\mathcal{N}_i| \rceil$ edges with lowest cosine similarity for every node $i$.
3: **for** epoch $= 1, 2, 3, \cdots$ **do**
4:     Obtain augmented graph views $\mathcal{G}^{hom}, \tilde{\mathcal{G}}^{hom}, \mathcal{G}^{het}, \tilde{\mathcal{G}}^{het}$ via random perturbations.
5:     Generate high-pass node representations $\boldsymbol{H}_H, \tilde{\boldsymbol{H}}_H$ based on Eq. (8), (9), using encoder weights $\boldsymbol{W}$.
6:     Generate low-pass node representations based on $\boldsymbol{H}_L, \tilde{\boldsymbol{H}}_L$ based on Eq. (6), (7)using encoder weights $\boldsymbol{W}$.
7:     Compute the contrastive objective $\mathcal{L}_{HLCL}$ in Eq. (12).
8:     Update the encoder weights $\boldsymbol{W}$ by applying stochastic gradient ascent to minimize $\mathcal{L}_{HLCL}$.
9:     **if** epoch $\% T = 0$ **then**
10:         update $\mathcal{G}^{het}, \mathcal{G}^{hom}, \tilde{\mathcal{G}}^{het}, \tilde{\mathcal{G}}^{hom}$ based on current node representations.
11:     **end if**
12: **end for**

---

all the nodes in the first augmented subgraph view. Since two views are symmetric, the loss for using the other view as anchor is defined in a similar fashion. The overall objective to be minimized is then defined as the average over all the four contrastive losses. Formally, we minimize:

$$\mathcal{L}_{\mathrm{HLCL}} = -\frac{1}{4N} \sum_{i=1}^{N} [l(\boldsymbol{z}_l^i, \tilde{\boldsymbol{z}}_l^i) + l(\boldsymbol{z}_h^i, \tilde{\boldsymbol{z}}_h^i) + l(\tilde{\boldsymbol{z}}_l^i, \boldsymbol{z}_l^i) + l(\tilde{\boldsymbol{z}}_h^i, \boldsymbol{z}_h^i)].$$
$$(12)$$

Effectively, by maximizing the agreement between the low-pass views and between the high-pass views, HLCL pulls away the representation of nodes with different features from their neighborhood, and allows them to be distinguished from their neighbors.

**Final representations.** After minimizing the contrastive loss in Eq. (12), we use the low-pass filtered representations as the final output.

The pseudocode is illustrated in Alg. 1.

**Scalability to Large Graphs via Message Passing.** The high-pass and low-pass filtered representations can be obtained through message passing in an inductive manner, according to Eq. (3), (4), without the need to explicitly calculate the normalized Adjacency and Laplacian matrix. In particular, the high-pass filtered representations can be obtained by iteratively differentiating the representations of a node and those of its neighbors, and the low-pass filtered representations can be obtained by aggregating the node's

representation with those of its neighbors:

$$h_i^l = \sigma(W^{l-1}h_i^{l-1}), \qquad (13)$$

$$(h_i^l)_L = \Sigma_{j \in \{\mathcal{N}_i^{hom} \cup \{i\}\}}(h_i^l + h_j^l), \qquad (14)$$

$$(h_i^l)_H = \Sigma_{j \in \{\mathcal{N}_i^{het} \cup \{i\}\}}(h_i^l - h_j^l). \qquad (15)$$

The above update rules can be applied to both augmented subgraphs. This is the same approach used to train GNNs on large graphs. Hence, HLCL will have the same complexity as conducting a normal GNN message passing with an additional message being passed to generate the high-pass filtered views. This makes HLCL scalable to large graphs, as we will also confirm in our experiments.

In addition, we will empirically confirm in Appendix A.4 that directly contrasting the high-pass and low-pass filtered representations can produce comparable results to HLCL, while speeding up the algorithm by 2x, as it requires minimizing only one pair of contrastive losses.

## 4.2 THEORETICAL ANALYSIS

Next, we theoretically prove that by separating the graph into homophilic and heterophilic subgraphs and applying low-pass and high-pass filters on them respectively, HLCL can encode both low-frequency and high-frequency information in the learned representations.

Following [Liu et al., 2022], we simplify the contrastive losses (10), (11) by assuming $\tau = 1$ and using inner product for $sim$. Additionally, we assume a one-layer linear encoder.

**Theorem 1** (HLCL: Spectral Invariance). *Under the above assumptions and given ideal subgraphs $G_{hom}$ and $G_{het}$, the HLCL loss can be lower-bounded as follows:*

$$\mathcal{L}_{\text{HLCL}} \geq \frac{-1-N}{2}\sum_i \Big(\alpha_{A_i}\big(2 - (\lambda_{A_i^{hom}} - \lambda_{\bar{A}_i^{hom}})^2\big)$$
$$+ \alpha_{L_i}\big(4 - (\lambda_{L_i^{het}} - \lambda_{\bar{L}_i^{het}})^2\big)\Big),$$

*where $\lambda_{A^{hom}}, \lambda_{\bar{A}_i^{hom}}$ denote the eigenvalues of the low-pass filters corresponding to augmented homophilic subgraph, $\lambda_{L^{het}}, \lambda_{\bar{L}_i^{het}}$ denote the eigenvalues of the high-pass filters corresponding to augmented heterophilic subgraph, and $\alpha_{A^{hom}}, \alpha_{L^{het}}$ are adaptive weights that change during the training as the parameters of the encoder changes.*

Theorem 1 provides a lower-bound for the HLCL loss. The lower-bound is in the form of a summation of two terms: the first term is the sum of the difference between the low-frequency components of the two low-pass filtered augmented views of the homophilic subgraph, and the second term is the sum of the difference between the two high-pass filtered augmented views of the heterophilic subgraph. Minimizing the HLCL loss ensures a small value for the lower bound. In doing so, the encoder changes such that it

assigns a larger weight ($\alpha_{A_i}$ and $\alpha_{L_i}$) to invariant frequencies $i$, for which $\lambda_{A_i}^{hom} \sim \hat{\lambda}_{A_i}^{hom}$ and $\lambda_{L_i}^{het} \sim \hat{\lambda}_{L_i}^{het}$. Notably, $(\lambda_{A_i}^{hom} \sim \lambda_{\bar{A}_i}^{hom})$ implies that the two contrasted augmentations are invariant at $i^{th}$ frequency. Same reasoning holds for the second term. Therefore, during training with HLCL, the encoder will emphasize the invariance between two contrasted augmentations from the spectrum domain, for both the homophilic and heterophilic subgraphs.

The proof is given in the Appendix. B

## 5 EXPERIMENTS

In this section, we evaluate the node representations learned with HLCL, under linear probe. We compare HLCL with existing graph CL, graph SSL and graph supervised learning methods, and conduct an extensive ablation study to evaluate the effect of each of HLCL's components.

**Datasets.** We consider nine widely-used public benchmark datasets with different homophily ratios, $\beta$. The details of the datasets are shown in Sec. A.2. We repeat the experiments 10 times for smaller benchmark datasets, and 3 times for large real-world datasets, and report the early-stopped average accuracy as the final result. For small graphs, we follow CPGNN [Zhu et al., 2020a], GRACE [Zhu et al., 2020c], and HGRL [Chen et al., 2022] and randomly select 10% of nodes for training, 10% of nodes for validation, and 80% of nodes for testing. For large graphs, following [Lim et al., 2021] we randomly select 25% of nodes for training, 25% of nodes for validation, and 50% of nodes for testing.

**Linear Probe Evaluation.** For SSL methods, we follow the evaluation protocol used in [Zhu et al., 2020c]. Models are first trained in a self-supervised manner without labels. Then, we fed the final node embeddings into a $l_2$-regularized logistic regression classifier to fit the labeled data.

### 5.1 RESULTS

**HLCL vs Self-supervised Baselines.** We compare HLCL with existing baselines for self-supervised representation learning. We consider general graph self-supervised learning methods like DGI [Velickovic et al., 2019], BGRL [Thakoor et al., 2021], and GRACE [Zhu et al., 2020c] as well as graph self-supervised learning methods that focus on learning under heterophily like HGRL [Chen et al., 2022], and SP-GCL [Wang et al., 2022]. In addition, we also include popular general graph supervised learning methods like GCN Kipf and Welling [2016], and graph supervised learning method targeting graphs under heterophily like MixHop, H2GCN, GloGNN, and CPGNN [Abu-El-Haija et al., 2019, Zhu et al., 2020b, 2021c, Li et al., 2022]. We record the hyperparameters for our experiments in Sec. A.1 Table 1 shows that HLCL achieves a significant boost on graphs with heterophily and a comparable performance

Table 1: HLCL vs baselines. Methods identified with † and ∗ are supervised methods and SSL methods for graphs under heterophily, respectively. HLCL achieves state-of-the-art under heterophily, and a comparable performance under homophily.

| | Homophily | | | Heterophily | | | | | |
| --- | --- | --- | --- | --- | --- | --- | --- | --- | --- |
| | Cora | CiteSeer | Pubmed | Actor | Chameleon | Squirrel | Penn94 | Twitch-gamers | Genius |
| Hom.($\beta$) | .83 | .71 | .79 | .09 | .23 | .19 | .48 | .56 | .51 |
| Nodes | 2,708 | 3,327 | 19,717 | 5,201 | 2,277 | 5,201 | 41,554 | 168,114 | 421,961 |
| Edges | 5,278 | 4,676 | 44,324 | 198,493 | 8,854 | 46,998 | 1,362,229 | 6,797,557 | 984,979 |
| Classes | 6 | 7 | 3 | 5 | 5 | 5 | 2 | 2 | 2 |
| **HLCL** | 84.1 ± 1.0 | 70.1 ± 0.8 | 84.5 ± 0.4 | 34.0 ± 0.2 | **50.9** ± 1.0 | **42.9** ± 2.6 | **68.1** ± 3.5 | **67.0** ± 0.9 | 84.3±0.1 |
| DGI | **84.5** ± 1.1 | **71.9** ± 0.7 | 86.0 ± 0.1 | 28.0 ± 1.4 | 32.6 ± 2.9 | 38.8 ± 2.3 | 62.9 ± 0.4 | 61.5 ± 0.6 | OOM |
| BGRL | 83.0 ± 0.7 | 69.8 ± 0.6 | 80.2 ± 0.6 | 28.3 ± 0.9 | 32.6 ± 4.7 | 35.7 ± 1.4 | 58.8 ± 0.6 | 60.9 ± 0.3 | 76.4±3.0 |
| GRACE | 83.7 ± 0.7 | 71.4 ± 1.0 | 86.7±0.1 | 34.5 ± 1.1 | 35.4 ± 3.6 | 36.2 ± 2.8 | 62.5 ± 0.4 | 57.1 ± 0.1 | 79.6±2.9 |
| SP-GCL* | 83.2 ± 0.1 | 72.0 ± 0.4 | 79.2 ± 0.7 | 27.7 ± 0.7 | 36.5 ± 1.9 | 33.7 ± 1.3 | - | 62.0 ± 0.2 | **90.1**\*± 0.2 |
| HGRL* | 82.1 ± 0.8 | 71.0 ± 0.7 | 84.2 ± 0.2 | 35.4 ± 0.9 | 43.9 ± 1.7 | 38.7 ± 1.7 | OOM | OOM | OOM |
| GCN† | 82.3 ± 1.2 | 70.2 ± 0.9 | 86.4 ± 0.3 | 28.2 ± 0.4 | 40.9 ± 4.1 | 39.5 ± 1.5 | 82.5 ± 0.3 | 62.2 ± 0.3 | 87.4 ± 0.4 |
| MixHop† | 81.0 ± 1.6 | 66.4 ± 1.7 | 85.1 ± 0.3 | 29.0 ± 1.0 | 33.8 ± 1.2 | 33.4 ± 1.6 | 83.5 ± 0.7 | 65.6 ± 0.3 | 90.6 ± 0.2 |
| H2GCN† | 81.4 ± 1.2 | 71.8 ± 0.9 | 85.9 ± 0.4 | 33.6 ± 0.8 | 26.8 ± 3.6 | 35.1 ± 1.2 | OOM | OOM | OOM |
| GloGNN† | **88.3**†± 1.1 | **77.4**†± 1.7 | **89.6**†± 0.4 | **37.4**†± 0.8 | 25.9 ± 3.6 | 35.1 ± 1.2 | **85.6**†± 0.4 | 66.4 ± 0.3 | **90.7**†± 0.1 |
| CPGNN† | 83.6 ± 1.3 | 72.0 ± 0.5 | 86.7 ± 0.2 | 35.6 ± 0.9 | 33.0 ± 3.2 | 30.0 ± 2.0 | OOM | OOM | OOM |

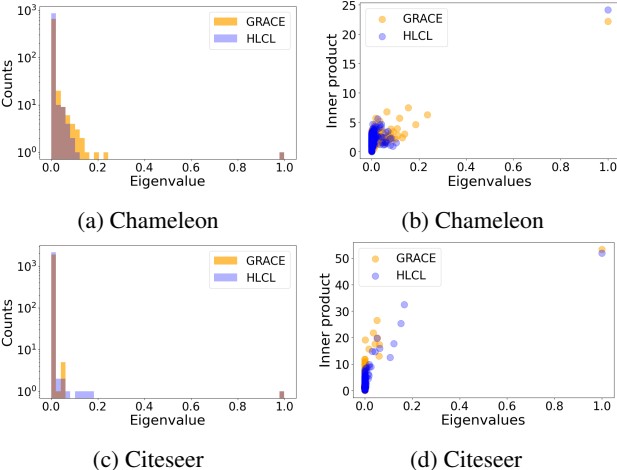

Figure 3: GRACE vs HLCL representations. (a), (c) distribution of eigenvalues in the representation matrix. (b), (d) alignment of the labels with the eigenvectors of the representation matrix. HLCL produces higher quality representations with lower rank and higher alignment with the label vector.

on graphs with homophily compared to the popular graph CL methods, showing up to 7% performance boost on Chameleon and 5% boost on Penn94. Compared to supervised methods such as H2GCN trained in an end-to-end manner, HLCL achieves a comparable performances under homophily and superior performance on heterophilic graphs like Chameleon by 10% and Squirrel by 3%. This confirms the effectiveness of HLCL.

**HLCL learns superior representations under heterophily.** Next, we compare the quality of representations learned by HLCL with that of GRACE, which only uses the low-pass filter for graph CL. We study Chameleon, a popular heterophily dataset [Platonov et al., 2023], and Citeseer,

a well-known homophily dataset [Yang et al., 2016]. Fig. 3 compares the distribution of normalized eigenvalues of the representation matrices and the alignment of their eigenvectors with the label vector. Lower-ranked representations that have a higher alignment between their prominent eigenvalues and label vector yield superior performance [Xue et al., 2022]. Fig. 3a confirms that HLCL's representations of Chameleon (heterophily) have a lower-rank structure compare to that of GRACE. Fig. 3b further confirms a strong alignment between the eigenvectors of the representation matrix and the label vector. Both factors contribute to higher classification accuracy of HLCL compared to GRACE. On the other hand, on Citeseer (homophily), HLCL exhibits a similar but only slightly higher rank spectrum than GRACE. Thus, contrasting the low-pass and high-pass views does not significantly harm the performances under homophily.

## 6 ABLATION STUDIES

### 6.1 HLCL WITH EXPLICIT AUGMENTATION

Graph augmentation methods are well studied for graph CL under homophily [Zhu et al., 2020c, You et al., 2020, Liu et al., 2022]. However, it is unclear if the same techniques are effective when a high-pass filter is applied to the graph. Here, we study the effects of different structural and feature augmentations applied to the heterophilic subgraphs of HLCL. We keep the graph augmentations on the homophilic subgraph constant, as mentioned in Sec. 4.1 and only investigate the effect of augmentation on the heterophilic subgraph. We consider popular graph augmentation methods including edge dropping, feature masking, node dropping, edge adding, and diffusion [You et al., 2020, Hassani and Khasahmadi, 2020]. As shown in Table 2, applying node dropping is

Table 2: The effect of applying different augmentations to the heterophilic subgraph.

| | Homophily | | Heterophily | | |
|---|---|---|---|---|---|
| | Cora | CiteSeer | Chameleon | Squirrel | Actor |
| EdgeRemoving | $80.0 \pm 0.5$ | $65.8 \pm 0.3$ | $47.6 \pm 1.1$ | $40.5 \pm 1.1$ | $32.4 \pm 0.9$ |
| NodeDropping | $\mathbf{82.1} \pm 0.5$ | $\mathbf{66.7} \pm 0.8$ | $48.0 \pm 3.4$ | $42.0 \pm 0.4$ | $32.9 \pm 0.5$ |
| EdgeAdding | $81.5 \pm 1.7$ | $66.2 \pm 0.8$ | $49.1 \pm 0.5$ | $\mathbf{42.9} \pm 2.6$ | $32.8 \pm 1.1$ |
| FeatureMasking | $81.9 \pm 1.8$ | $65.6 \pm 1.5$ | $48.3 \pm 1.8$ | $40.8 \pm 1.3$ | $\mathbf{33.8} \pm 1.5$ |
| PPRDiffusion | $75.1 \pm 1.8$ | $62.0 \pm 1.3$ | $\mathbf{50.2} \pm 4.7$ | $40.7 \pm 0.2$ | $33.2 \pm 1.8$ |

Table 3: Using high-pass (HP) only, low-pass only (LP) filter or both filters (HLCL) with inferred and ideal homophilic and heterophilic subgraphs (found using actual labels).

| | Homophily | Heterophily | |
|---|---|---|---|
| | Cora | Chameleon | Squirrel |
| **HLCL** | **84.1** | **50.9** | **42.9** |
| LP | 83.7 | 35.4 | 36.2 |
| HP | 32.5 | 33.1 | 33.1 |
| HLCL:ideal | 89.7 | 61.6 | 47.4 |
| LP:ideal | 87.1 | 53.7 | 44.9 |
| HP:ideal | 63.6 | 58.9 | 39.9 |

Table 4: Performance for different update intervals.

| Dataset | $T = 10$ | $T = 50$ | $T = 250$ | No Update |
|---|---|---|---|---|
| Cora | 80.5 | **84.1** | 83.1 | 82.1 |
| Chameleon | 41.6 | **50.9** | 48.3 | 42.7 |

more effective on improving the performance on homophilic graph, while feature perturbation is more effective on improving the performance on heterophilic graph. Overall, HLCL's performance is stable across all augmentations.

## 6.2 HLCL WITH SINGLE GRAPH FILTER

Next, to confirm that both filters are necessary for HLCL's superior performance, we examine the performance of HLCL while applying contrastive loss to either low-pass or high-pass filtered representations during training. The results are shown in Table 3. To rule out the possibility for poor subgraph sampling influencing the results, we also consider ideal subgraphs obtained via the true labels. That is, in $\mathcal{G}^{hom}$, only nodes of the same labels are connected, while in $\mathcal{G}^{het}$, only nodes of different labels are connected. First, we observe that applying contrastive loss to both high-pass and low-pass filtered representations yields the best performance, both on regular subgraphs and ideal subgraphs. This demonstrate that utilizing both representations from both frequency terms is crucial for HLCL's success. Besides, we see that more precise homophilic and heterophilic subgraphs considerably improves the performance, and finding them more accurately is a promising direction for future work.

In Appendix A.3, we demonstrate the performance of existing graph CL methods using only high-pass filters.

## 6.3 SUBGRAPH UPDATE INTERVAL

We also conduct an ablation study on the interval between subgraph updates. The results are shown in Table 4. We see that frequent updates ($T = 10$) or no updates can both harm the performance on both homophilic and heterophilic graphs. Not updating the subgraphs leads to overfitting their inaccuracies, and updating them too frequently does not allow aggregating and learning the information effectively. A moderate amount of updates yields best performance.

## 6.4 HOMOPHILY RATIO CAN GUIDE TUNING

Next, we conduct a detailed study to explore the impact of varying $k_1$ on HLCL's performance. As observed in Table 5, different $k_1$ (and $k_2 = 1 - k_1$) values can have a significant influence on the performance. The performance on the Cora dataset varies by 30% with different $k_1$ values, while the performance of the Chameleon dataset varies by 7%. Based on the results, the homophily ratio of the graph is a good indicator of the appropriate $k_1$ (and $k_2$) values. On both Cora and Chameleon, the best performances are achieved when $k_1 \approx$ homophily ratio. Similar results are observed in Citeseer ($k_1 = 0.9$; homophily ratio = 0.71) and Actor ($k_1 = 0.09$, homophily ratio = 0.09). In practice, one can sample a small subgraph and measure its homophily ratio for easier tuning.

Table 5: Performance for different values of $k_1$.

| Dataset | 0.9 | 0.8 | 0.5 | 0.2 | 0.1 |
|---|---|---|---|---|---|
| Cora ($\beta = 83$) | **84.1** | 80.2 | 72.1 | 54.8 | 53.7 |
| Chameleon ($\beta = 23$) | 42.0 | 42.0 | 42.0 | **50.9** | 45.0 |

## 6.5 HLCL SUBGRAPH INFERENCE

We also investigate the connectivity of the homophilic and heterophilic subgraphs inferred by HLCL. Specifically, we measured the fraction of nodes in the largest connected component of the original graph that are in the largest connected

Table 6: Connectivity of the inferred homophilic and heterophilic subgraphs.

| Data | homophilic | heterophilic |
|------|------------|--------------|
| Cora (.83) | 1 | 8.5% |
| Citeseer (.71) | 1 | 7.7% |
| Chameleon (.23) | 25.6% | 98.2% |
| Squirrel (.19) | 94.4% | 95% |
| Actor (.09) | 4.7% | 99.8% |

Table 7: Producing final representations with different graph filters. Low-pass filtered representations has the highest performance on both homophilic and heterophilic graphs.

| Model | Cora | Citeseer | Chameleon | Squirrel |
|-------|------|----------|-----------|----------|
| LP | **84.1** $\pm$ 1.0 | **70.1** $\pm$ 0.8 | **50.9** $\pm$ 1.0 | **42.9** $\pm$ 2.6 |
| HP | 51.9 $\pm$ 2.9 | 36.2 $\pm$ 2.5 | 35.6 $\pm$ 3.1 | 29.8 $\pm$ 1.6 |
| LP+HP | 74.2 $\pm$ 2.0 | 57.6 $\pm$ 2.0 | 48.7 $\pm$ 1.0 | 40.8 $\pm$ 2.0 |

component of each subgraph after sampling in Table 6. We see that under homophily (Cora, Citeseer), all the nodes are in the homophilic subgraphs and the heterophilic subgraph is small and minimally affects the performance. Under extreme heterophily (Actor) almost all the nodes are in the heterophilic subgraph and the homophilic subgraph is small and minimally affects the performance. For other graphs, depending on the tuned value of $k_1$, the size of the largest connected component of the two subgraphs changes.

## 6.6 USING DIFFERENT FILTERED REPRESENTATIONS AS OUTPUT

Finally, we study the performance of using different filters to produce final representations. We consider using low-pass filtered only, high-pass filtered only, and concatenating the low-pass filtered and high-pass filtered representations. The results are shown in Table 7. We observe that using low-pass filtered representations can yield better performances for both homophilic and heterophilic graphs. It is important to note that the encoder is trained using contrastive loss on both high-pass filtered heterophilic subgraphs and low-pass filtered homophilic subgraphs. This ensures that nodes in the same class have similar representations when a low-pass filter is applied, and nodes in different classes have distinct representations with a high-pass filter. During inference, the goal is to identify nodes with similar representations. Hence, using low-pass filtered representatives work better in practice, than using the high-pass filtered representations or a combination of both.

## 7 LIMITATIONS

The performance of learning from heterophily graphs heavily depends on how information is aggregated in different neighborhoods. Label availability is crucial for guiding this aggregation. In the absence of labels, a significant challenge for any graph SSL method, including HLCL, arises when nodes with similar labels cannot be approximately identified. For instance, in the Penn94 dataset, all SSL methods in Table 1, including HLCL, underperform compared to supervised methods. This is due to the lack of correlation between node feature similarity and label similarity in this dataset. In contrast, in the Cora dataset, nodes belonging to the same

class exhibit an average of 31% higher feature cosine similarity than nodes from different classes, while in Chameleon, this difference is 11%. However, in Penn94, the difference is only 1.4% on average, indicating a high similarity in node features across different classes. Consequently, SSL methods, including HLCL, face challenges in learning high-quality node representations. Despite this, HLCL outperforms other SSL baselines on Penn94, as shown in Table 1.

Additionally, on graphs where node features cannot differentiate different classes, HLCL can face challenges. For instance, for Actor dataset, nodes from different classes have similar connectivity patterns to other classes [Ma et al.]. In this case, the graph structure is not useful to classify the nodes (and may hurt the performance), and only relying on node features achieve a better performance. As shown in [Ma et al., Chen et al., 2022], models like MLP which do not use any graph structure can outperform GNN methods like GCN and even H2GCN on the Actor dataset. HGRL uses MLP as its encoder and does not leverage the graph structure, hence it can slightly outperform HLCL on Actor, but is outperformed by HLCL on other datasets.

In Sec. A.5, we discuss the effectiveness of node features in inferring subgraphs, and their limitations if used directly for classification without incorporating the graph structure.

## 8 CONCLUSION

We proposed HLCL, a contrastive learning framework that finds a homophilic and a heterophilic subgraph in a graph, applies high-pass and low-pass filters to the augmented subgraph views, and learns node representations by contrasting the filtered augmented views. This is particularly beneficial for graphs with heterophily. Through extensive experiments, we demonstrated that our proposed framework achieves up to 7% boost graphs under heterophily and outperforms popular graph supervised learning methods by up to 10%. HLCL also provides a comparable performance under homophily. We believe our work provides an important direction for future work on contrastive learning under heterophily.

**Acknowledgments.** This research was partially supported by the National Science Foundation CAREER Award 2146492.

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

# Graph Contrastive Learning under Heterophily via Graph Filters (Supplementary Material)

**Wenhan Yang**[1]                    **Baharan Mirzasoleiman**[1]

[1]Computer Science Department, University of California Los Angeles (UCLA)

## A    APPENDIX

### A.1    HYPERPARAMETERS DETAILS

We list the details of our model hyperparameters for each datasets in Table. 8.

Table 8: HLCL hyperparameters for each dataset

|       | Cora | CiteS | Pubmed | Actor | Cham | Squir | Penn | TwitchG | Genius |
|-------|------|-------|--------|-------|------|-------|------|---------|--------|
| $k_2$ | 0.9  | 0.9   | 0.8    | 0.1   | 0.2  | 0.5   | 1.0  | 1.0     | 1.0    |
| $k_1$ | 0.1  | 0.1   | 0.2    | 0.9   | 0.8  | 0.5   | 1.0  | 1.0     | 1.0    |
| lr    | 1e-3 | 1e-3  | 1e-3   | 1e-3  | 1e-3 | 1e-3  | 1e-3 | 1e-3    | 1e-3   |
| $T$   | 50   | 50    | 50     | 50    | 50   | 50    | 50   | 50      | 50     |

### A.2    DATASET DETAILS

For graphs with homophily, we use the citation networks including Cora, Citeseer, and Pubmed [Yang et al., 2016]. For graphs with heterophily, we use the Wikipedia network and the web page networks including Chameleon, Squirrel, and Actor [Rozemberczki et al., 2021, Pei et al., 2020]. Note that, for fair comparison, we adopt the Chameleon and Squirrel from [Platonov et al., 2023] with duplicated nodes removed. To illustrate the scalability of HLCL, we also include three large-scale real-world datasets, Penn94, Genius, and Twitch-gamers provided by [Lim et al., 2021].

### A.3    EXISTING GCL METHODS WITH HP FILTERS

In this section, we illustrate the importance of our contrastive structure in achieving performance gains on heterophily datasets. We show that contrasting both the low-pass filtered graph views and high-pass filtered graph views is crucial to obtain high-quality representation under heterophily, as opposed to applying high-pass filter. To do so, we replace the LP filter with HP filter in other popular graph CL methods. The results are shown in Table 9. As demonstrated, while there are performance gains on some heterophily datasets, accuracy significantly deteriorates in homophily settings. For larger values of $\beta$, it is more likely that nodes with the same labels are connected together. In graphs with a large homophily ratio, most of the neighborhoods have homogeneous labels. On the other hand, graphs with a small homophily ratio contain neighborhoods with homogeneous and heterogeneous labels, as illustrated in Fig. 2. Existing graph CL methods have a very poor performance under heterophily, or low homophily ratio, and cannot learn high-quality representations.

*Accepted for the 40[th] Conference on Uncertainty in Artificial Intelligence* (UAI 2024).

Table 9: Using high-pass filter in existing Graph CL methods. HLCL denotes our method

| | Homophily | | Heterophily | | |
|---|---|---|---|---|---|
| | Cora | CiteSeer | Chameleon | Squirrel | Actor |
| **HLCL** | $84.1 \pm 1.0$ | $70.1 \pm 0.8$ | $50.9 \pm 1.0$ | $42.9 \pm 2.6$ | $34.0 \pm 0.2$ |
| DGI:low | $84.53 \pm 1.1$ | $71.88 \pm 0.7$ | $32.58 \pm 2.9$ | $38.83 \pm 2.3$ | $28.00 \pm 1.4$ |
| DGI:high | $31.95 \pm 2.8$ | $30.54 \pm 1.7$ | $29.89 \pm 3.0$ | $36.86 \pm 3.0$ | $32.03 \pm 0.9$ |
| BGRL:low | $83.01 \pm 0.7$ | $69.81 \pm 0.6$ | $32.58 \pm 4.7$ | $35.70 \pm 1.4$ | $28.32 \pm 0.9$ |
| BGRL:high | $29.63 \pm 2.8$ | $24.99 \pm 3.1$ | $37.30 \pm 6.0$ | $38.03 \pm 1.1$ | $33.87 \pm 1.9$ |
| GRACE:low | $83.69 \pm 0.7$ | $71.37 \pm 1.0$ | $35.39 \pm 3.6$ | $36.18 \pm 2.8$ | $34.5 \pm 1.1$ |
| GRACE:high | $32.46 \pm 2.0$ | $26.55 \pm 3.1$ | $33.03 \pm 3.9$ | $33.05 \pm 2.1$ | $32.00 \pm 1.3$ |

## A.4 SIMPLIFIED HLCL

Empirically, we observed that directly contrasting the high-pass filtered representations with the low-pass filtered representations can produce comparable results to HLCL, as shown in Table 10. This simplified version can speed up the algorithm by $2\times$, as it requires only one contrasting learning process.

Table 10: Comprasion between HLCL and Simplified HLCL

| | Homophily | | Heterophily | | |
|---|---|---|---|---|---|
| | Cora | CiteSeer | Chameleon | Squirrel | Actor |
| **HLCL** | $84.1 \pm 1.0$ | $70.1 \pm 0.8$ | $50.9 \pm 1.0$ | $42.9 \pm 2.6$ | $34.0 \pm 0.2$ |
| HLCL$_{Simplified}$ | $83.5 \pm 2.7$ | $71.8 \pm 1.4$ | $48.3 \pm 6.8$ | $39.5 \pm 5.3$ | $35.5 \pm 1.9$ |

## A.5 USING FEATURES TO INFER LABEL INFORMATION

HLCL uses feature information to approximately estimate the label information. Here, we justify this choice empirically and demonstrate that while feature information can help in inferring subgraphs approximately, it cannot be used for accurte node classification. First, we show that the node features are sufficient to give approximate neighborhood information, which is helpful in splitting the subgraph. We provide the homophily ratios of the original graph, the homophilic subgraph, and the heterophilic subgraph selected based on feature similarity across different datasets. As shown in Table 11, using feature cosine similarity, HLCL can approximately create homophilic and heterophilic subgraphs from the original graph. However, while features can approximately indicate if neighboring nodes are of the same class, they are insufficient for accurate (multi-class) node classification, and the graph structure is crucial to take into account. Otherwise, one could simply use an MLP classifier on node features. It is important to note that approximately identifying a homophilic and a heterophilic subgraph is a binary classification task, which is significantly easier than multi-class node classification. We show the insufficiency of node features for accurate classification without graph structure in Table 12. We conducted additional experiments with an MLP classifier on various homophily and heterophily datasets, which showed that MLP yields very poor performances, particularly under heterophily.

Table 11: Homophily ratios of the subgraphs sampled via node features. After sampling, homophilic subgraph has a higher homophily ratio, while the heterophilic subgraph has a lower homophily ratio compared to the original graph.

| | Cora(hom) | Citeseer(hom) | Chameleon(het) | Squirrel(het) |
|---|---|---|---|---|
| **orig graph hom%** | 0.83 | 0.71 | 0.23 | 0.19 |
| **hom subgraph hom%** | 0.87 | 0.82 | 0.74 | 0.42 |
| **het subgraph hom%** | 0.08 | 0.05 | 0.24 | 0.19 |

Table 12: Using node feature only (MLP) to classify the nodes. As shown, without graph structures, the model can only achieve sub-optimal performances.

|       | Cora (6 classes) | Citeseer (7 classes) | Chameleon (5 classes) | Squirrel (5 classes) |
|-------|------------------|----------------------|-----------------------|----------------------|
| MLP   | $64.8 \pm 1.2$   | $66.5 \pm 1.0$       | $37.4 \pm 2.1$        | $25.5 \pm 0.9$       |
| HLCL  | $84.1 \pm 1.0$   | $70.1 \pm 0.8$       | $50.9 \pm 1.0$        | $42.9 \pm 2.6$       |

## A.6 EXTENDED RELATED WORK

**(Semi-)supervised learning on graphs.** In recent years, GNNs have become one of the most prominent tools for processing graph-structured data. In general, GNNs utilize the adjacency matrix to learn the node representations, by aggregating information within every node's neighborhood [Defferrard et al., 2016, Kipf and Welling, 2016]. Existing variants, including GraphSAGE [Hamilton et al., 2017], Graph Attention (GAT) [Veličković et al., 2017], MixHop [Abu-El-Haija et al., 2019], SGC [Nt and Maehara, 2019], GAT [Velickovic et al., 2019], and GIN [Xu et al., 2018], learn a more general class of neighborhood mixing relationships, by aggregating weighted information within a multi-hop neighborhood of every node. GNNs can be generally seen as applying a fix, or a parametric and learnable (e.g. GAT) low-pass graph filter to graph signals. Those with trainable parameters can adapt to a wider range of frequency levels on different graphs. However, they still have a higher emphasis on lower-frequency signals and discard the high-frequency signals in a graph. While the aggregation operation makes GNNs powerful tools for semi-supervised learning, it can make the learned node representations indistinguishable in a neighborhood [Nt and Maehara, 2019]. As a result, typical GNNs and their variants have been long criticized for their poor generalization performance under heterophily [Balcilar et al., 2020].

**Graph self-supervised learning.** Graph self-supervised learning methods have become a powerful tool for learning representations without any labels, and graph contrastive learning is the most successful and popular model structure. Numerous methods have been proposed in the field: [Velickovic et al., 2019, Peng et al., 2020, Hassani and Khasahmadi, 2020, Zhu et al., 2021c] focus on contrasting the global augmented representation with the local augmented representation, while [Zhu et al., 2020c, You et al., 2020, Qiu et al., 2020, Liu et al., 2022] contrast same-scale representation, global or local, in two augmented views. Due to the complexity of collecting negative samples in graph data, negative-sample free contrastive objectives have also been studied [Thakoor et al., 2021, Bielak et al., 2021]. However, works mentioned above focus on encoding the homophily graphs and perform poorly on graphs with heterophily. Recently, a stream of self-supervised learning methods have been proposed to learn effectively the node representations of the heterophily graphs without any labels. HGRL [Chen et al., 2022] improves the node representations on heterophilic graphs by preserving the node original features and rewiring informative nodes that are not in the local neighborhood. SP-GCL [Wang et al., 2022] proposed using nodes from the T-hop neighborhood of a node with high feature similarities as positive pairs, without using any explicit augmentations. DSSL [Xiao et al., 2022] separates the heterogeneous patterns in local neighborhood distributions to capture both homophilic and heterophilic information globally. GREET [Liu et al., 2023] discriminates homophilic edges from heterophilic edges using random walk based graph diffusion and contrasts the projected representations of the two graph views directly via a dual-channel contrastive loss. MUSE [Yuan et al., 2023] utilize semantic view contrast based on ego node feature perturbations and contextual view contrast based on topology perturbations. Then, it integrates the representations learned from both contrasting views to construct a fusion contrast that combines both structural and semantic information. NeCo [He et al., 2023] proposes a new pretext task, group discrimination, which divides the nodes into k groups and keeps the consistent representation of nodes within a group.

**Graph (semi-)supervised learning under heterophily.** To address over-smoothing issue of GNNs, recent methods propose to use other types of aggregation that better fit graphs with heterophily. Geom-GCN uses geometric aggregation in place of the typical aggregation [Pei et al., 2020], $H_2$GCN uses several special model designs including separate aggregation and higher-neighborhood aggregation to train the model for handling graphs with heterophily, and CPGNN trains a compatibility matrix to model the heterophily level [Zhu et al., 2020a]. More recently, Wang et al. [2019] proposed to learn an aggregation filter for every graph from a set of based filters designed based on different ways of normalizing the adjacency matrix. GGCN introduced degree corrections and signed message passing on GCN to address both oversmoothing problems and the model's poor performances on heterophily graphs [Yan et al., 2021]. Zhu et al. [2021a] analyzed and designed a uniform framework for GNNs propagations and proposed GNN-LF and GNN-HF that preserve information of different frequency separately by using different filtering kernels with learnable weights. FAGCN [Bo et al., 2021] and FBGNN [Luan et al., 2020] train two *separate* encoders to capture the high-pass and low-pass graph signals separately. Then they rely on labels to learn relatively complex mechanisms to combine the outputs of the encoders. However, learning how to combine the encoder outputs is highly sensitive to having high-quality labels. This makes such methods highly impractical for unsupervised

contrastive learning, where the label information is not available.

Unlike the above supervised methods, we apply the high-pass and low-pass filters to different subgraphs, contrasting the resulting high-pass filtered node views and low-pass filtered node views in a self-supervised manner, without any label. This is in contrast to learning the best combination of filtered signals of different encoders based on labels.

## B   PROOF

**Assumption 1.** *Let $X$ be the feature matrix of $\mathcal{G}^{hom}$ and $W$ be the learnable weights of the GNN encoder. Then,*

$$XWWX = w_0 + w_1 A^{hom} + w_2 (A^{hom})^2 + \cdots + w_j (A^{hom})^j.$$

$XWWX$ under homophily captures the similarities of features between every two nodes in the subgraph after passing through the low-pass graph filter. Assumptions 1 aims to expand $XWWX$ with the weighted sum of different orders of $A$. Here, $w_i$ s are the weights of different orders of $A$. That is $w_i$ is the weight of $i$-th order of $A$, representing the number of length-$i$ paths between nodes $i$ and $j$ in its $(i, j)$ entry. For homophilic subgraphs, which adhere to the homophily principle, the weights for closer-hop connections (represented by $A$, $A^2$, etc.) are higher, since the closer the nodes are, the more similar they are. This is based on the homophily principle [McPherson et al., 2001, Luan et al., 2020]. This principle suggests that, in homophily graphs, nodes within closer neighborhoods exhibit greater feature similarities. After projection, the similarities also become higher [Zhang et al., 2018].

**Assumption 2.** *Let $X$ be the feature matrix of $\mathcal{G}^{het}$ and $W$ be the learnable weights of the GNN encoder. Then,*

$$XWWX = w_0 + w_1 L^{het} + w_2 (L^{het})^2 + \cdots + w_j (L^{het})^j.$$

$XWWX$ under heterophily captures the dissimilarities of features between every two nodes in the subgraph after passing through the high-pass graph filters. Assumptions 2 aims to expand $XWWX$ with the weighted sum of different orders of $L$. Here, $w_i$ is the weight of $i$-th order of $L$. In contrast to homophilic graphs, for heterophilic subgraphs, the closer the nodes are, the more dissimilar they are [Zhu et al., 2020c].

**Lemma 1.** *Let $A$ and $\widetilde{A}$ be adjacency matrices of the target graph and its augmented counterpart. Suppose that $A$ and $\widetilde{A}$ have the same eigenspaces, and let $D$ and $\widetilde{D}$ be the corresponding degree matrices, where $D = \widetilde{D}$. Then the Laplacian matrices $L$ and $\widetilde{L}$ have the same eigenspaces.*

*Proof.*   Given that $A$ and $\widetilde{A}$ have the same eigenspaces, there exists an orthogonal matrix $Q$ such that:

$$A = Q \Lambda Q^T \quad \text{and} \quad \widetilde{A} = Q \widetilde{\Lambda} Q^T$$

where $\Lambda$ and $\widetilde{\Lambda}$ are diagonal matrices containing the eigenvalues of $A$ and $\widetilde{A}$, respectively. Since $D = \widetilde{D}$, let $D = \widetilde{D}$. The Laplacian matrices are defined as:

$$L = D - A \quad \text{and} \quad \widetilde{L} = D - \widetilde{A}$$

Substituting the spectral decompositions of $A$ and $\widetilde{A}$, we have:

$$L = D - Q \Lambda Q^T$$
$$\widetilde{L} = D - Q \widetilde{\Lambda} Q^T$$

Both $L$ and $\widetilde{L}$ can be written as:

$$L = Q(Q^T D Q - \Lambda) Q^T$$
$$\widetilde{L} = Q(Q^T D Q - \widetilde{\Lambda}) Q^T$$

Since $D$ is diagonal, $Q^T D Q$ remains a diagonal matrix (as the orthogonal transformation of a diagonal matrix preserves diagonal structure). Let $D' = Q^T D Q$, then:

$$L = Q(D' - \Lambda) Q^T$$
$$\widetilde{L} = Q(D' - \widetilde{\Lambda}) Q^T$$

The eigenvalues of $L$ and $\widetilde{L}$ are given by the diagonal entries of $D' - \Lambda$ and $D' - \widetilde{\Lambda}$, respectively. Since $Q$ is the same for both $L$ and $\widetilde{L}$, they have the same eigenspaces. Thus, $L$ and $\widetilde{L}$ have the same eigenspaces.   $\square$

## B.1 THEOREM 1 [HLCL: SPECTRAL INVARIANCE]

Given a graph $\mathcal{G}$, we infer a homophilic and a heterophilic subgraph from it, denoted as $\mathcal{G}_{\text{hom}}$ and $\mathcal{G}_{\text{het}}$, respectively. Their augmented counterparts are denoted as $\tilde{\mathcal{G}}_{\text{hom}}$ and $\tilde{\mathcal{G}}_{\text{het}}$. For graph augmentations, we follow [Liu et al., 2022], where the adjacency matrix of the homophilic subgraph and the augmented homophilic subgraph share the same eigenspaces ($\boldsymbol{A}_{\text{hom}}$ and $\tilde{\boldsymbol{A}}_{\text{hom}}$). Similarly, the adjacency matrix of the heterophilic subgraph and the augmented heterophilic subgraph share the same eigenspaces ($\boldsymbol{A}_{\text{het}}$ and $\tilde{\boldsymbol{A}}_{\text{het}}$). By Lemma 1, the Laplacian matrix of the homophilic subgraph and the augmented homophilic subgraph share the same eigenspaces ($\boldsymbol{L}_{\text{hom}}$ and $\tilde{\boldsymbol{L}}_{\text{hom}}$), and the Laplacian matrix of the heterophilic subgraph and the augmented heterophilic subgraph share the same eigenspaces ($\boldsymbol{L}_{\text{het}}$ and $\tilde{\boldsymbol{L}}_{\text{het}}$).

We establish the following lower bound:

$$\mathcal{L}_{\text{HLCL}} \geq \frac{-1-N}{2} \sum_i \left( \alpha_{\boldsymbol{A}_i} \left( 2 - (\lambda_{\boldsymbol{A}_i^{\text{hom}}} - \lambda_{\tilde{\boldsymbol{A}}_i^{\text{hom}}})^2 \right) + \alpha_{\boldsymbol{L}_i} \left( 4 - (\lambda_{\boldsymbol{L}_i^{\text{het}}} - \lambda_{\tilde{\boldsymbol{L}}_i^{\text{het}}})^2 \right) \right)$$

where $\lambda_{\boldsymbol{A}^{\text{hom}}}$ and $\lambda_{\boldsymbol{L}^{\text{het}}}$ denote the eigenvalues of the homophilic subgraph low-pass filter and the heterophilic subgraph high-pass filter, respectively, and $\alpha_{\boldsymbol{A}^{\text{hom}}}$ and $\alpha_{\boldsymbol{L}^{\text{het}}}$ denote the adaptive weights for the $i$-th adjacency and Laplacian matrix components.

*Proof.* By minimizing the HLCL loss, we minimize the losses for contrasting augmented views of both heterophilic and homophilc subgraphs. Hence we discuss each in our proof.

For simplification, since the HLCL loss is symmetric, we only choose one graph view as the anchor view.

$$\mathcal{L} = -\frac{1}{2N} \sum_{i=1}^{N} (l(\boldsymbol{z}_l^i, \tilde{\boldsymbol{z}}_l^i) + l(\boldsymbol{z}_h^i, \tilde{\boldsymbol{z}}_h^i)) \tag{16}$$

$$= -\frac{1}{2N} \sum_{i=1}^{N} (\log \frac{e^{\boldsymbol{z}_l^i \tilde{\boldsymbol{z}}_l^{i\,T}}}{e^{\boldsymbol{z}_h^i \tilde{\boldsymbol{z}}_h^{i\,T}} + \sum_{\substack{k \in [N],) \\ k \neq i}} e^{\boldsymbol{z}_l^i \tilde{\boldsymbol{z}}_l^{k\,T}}} + \log \frac{e^{\boldsymbol{z}_h^i \tilde{\boldsymbol{z}}_h^{i\,T}}}{e^{\boldsymbol{z}_h^i \tilde{\boldsymbol{z}}_h^{i\,T}} + \sum_{\substack{k \in [N], \\ k \neq i}} e^{\boldsymbol{z}_h^i \tilde{\boldsymbol{z}}_h^{k\,T}}}) \tag{17}$$

$$= -\frac{1}{2N} \sum_{i=1}^{N} (\boldsymbol{z}_l^i \tilde{\boldsymbol{z}}_l^{i\,T} + \boldsymbol{z}_h^i \tilde{\boldsymbol{z}}_h^{i\,T} - \log \sum_k^N e^{\boldsymbol{z}_l^i \tilde{\boldsymbol{z}}_l^{k\,T}} - \log \sum_k^N e^{\boldsymbol{z}_h^i \tilde{\boldsymbol{z}}_h^{k\,T}}) \tag{18}$$

$$\geq -\frac{1}{2N} \sum_{i=1}^{N} (\boldsymbol{z}_l^i \tilde{\boldsymbol{z}}_l^{i\,T} + \boldsymbol{z}_h^i \tilde{\boldsymbol{z}}_h^{i\,T} - \log N \cdot e^{\sum_k^N \boldsymbol{z}_l^i \tilde{\boldsymbol{z}}_l^{k\,T}/N} - \log N \cdot e^{\sum_k^N \boldsymbol{z}_h^i \tilde{\boldsymbol{z}}_h^{k\,T}/N}) \tag{19}$$

$$\equiv - \sum_{i=1}^{N} (\boldsymbol{z}_l^i \tilde{\boldsymbol{z}}_l^{i\,T} + \boldsymbol{z}_h^i \tilde{\boldsymbol{z}}_h^{i\,T} - \frac{1}{N} \sum_N \boldsymbol{z}_l^i \tilde{\boldsymbol{z}}_l^{i\,T} + \boldsymbol{z}_h^i \tilde{\boldsymbol{z}}_h^{i\,T}) \tag{20}$$

$$= -(tr(\boldsymbol{Z}_l \tilde{\boldsymbol{Z}}_l^T) + tr(\boldsymbol{Z}_h \tilde{\boldsymbol{Z}}_h^T) - \frac{1}{N} sum(\boldsymbol{Z}_l \tilde{\boldsymbol{Z}}_l^T) - \frac{1}{N} sum(\boldsymbol{Z}_h \tilde{\boldsymbol{Z}}_h^T)) \tag{21}$$

$\boldsymbol{Z}_l$ is the projected representation of $\mathcal{G}_{\text{hom}}$, $\tilde{\boldsymbol{Z}}_l$ is the projected representation of $\tilde{\mathcal{G}}_{\text{hom}}$, $\boldsymbol{Z}_h$ is the projected representation of $\mathcal{G}_{\text{het}}$, and $\tilde{\boldsymbol{Z}}_h$ is the projected representation of $\tilde{\mathcal{G}}_{\text{het}}$. As mentioned before, $\boldsymbol{A}_{\text{hom}}$ and $\tilde{\boldsymbol{A}}_{\text{hom}}$ share the same eigenspaces, so we have that $\boldsymbol{A}_{\text{hom}} = \boldsymbol{Q}_{\text{hom}} \Lambda_{\text{hom}} \boldsymbol{Q}_{\text{hom}}^T$ and $\tilde{\boldsymbol{A}}_{\text{hom}} = \boldsymbol{Q}_{\text{hom}} \tilde{\Lambda}_{\text{hom}} \boldsymbol{Q}_{\text{hom}}^T$, where $\boldsymbol{Q}_{\text{hom}}$ is the collection of eigenspaces, and $\Lambda_{\text{hom}} = \text{diag}(\lambda_{\boldsymbol{A}_i^{hom}})$ and $\tilde{\Lambda}_{\text{hom}} = \text{diag}(\lambda_{\tilde{\boldsymbol{A}}_i^{hom}})$ are their diagonal weight matrices. Similarly, $\boldsymbol{A}_{\text{het}} = \boldsymbol{Q}_{\text{het}} \Lambda_{\text{het}} \boldsymbol{Q}_{\text{het}}^T$ and $\tilde{\boldsymbol{A}}_{\text{het}} = \boldsymbol{Q}_{\text{het}} \tilde{\Lambda}_{\text{het}} \boldsymbol{Q}_{\text{het}}^T$, where $\boldsymbol{Q}_{\text{het}}$ is the collection of eigenspaces, and $\Lambda_{\text{het}} = \text{diag}(\lambda_{\boldsymbol{L}_i^{het}})$ and $\tilde{\Lambda}_{\text{het}} = \text{diag}(\lambda_{\tilde{\boldsymbol{L}}_i^{het}})$. With the simplification of the HLCL loss, we have $\boldsymbol{Z}_h \tilde{\boldsymbol{Z}}_h^T = \boldsymbol{L} \boldsymbol{X} \boldsymbol{W} \boldsymbol{W} \boldsymbol{X} \tilde{\boldsymbol{L}}$ and $\boldsymbol{Z}_l \tilde{\boldsymbol{Z}}_l^T = \boldsymbol{A} \boldsymbol{X} \boldsymbol{W} \boldsymbol{W} \boldsymbol{X} \tilde{\boldsymbol{A}}$, where $W$ is learnable parameters of the encoder.

**Lemma 2.** *With assumption 1, for homophilic subgraph $\mathcal{G}^{hom}$, when $j \geq N-1$, $\boldsymbol{X} \boldsymbol{W} \boldsymbol{W} \boldsymbol{X} = w_0 + w_1 \boldsymbol{A}^{hom} + w_2 (\boldsymbol{A}^{hom})^2 + \cdots + w_j (\boldsymbol{A}^{hom})^j = \boldsymbol{Q}_{hom} \boldsymbol{A}_{hom} \boldsymbol{Q}_{hom}^T$, where $\boldsymbol{A}_{hom} = diag(\alpha_{\boldsymbol{A}_1} \ldots \alpha_{\boldsymbol{A}_N})$. $\alpha_{\boldsymbol{A}_1} \ldots \alpha_{\boldsymbol{A}_N}$ are N different parameters, if $\lambda_{\boldsymbol{A}_1^{hom}} \ldots \lambda_{\boldsymbol{A}_N^{hom}}$ are N different frequency amplitudes.*

*Proof.* The proof can be found in Theorem 4 of [Liu et al., 2022]. □

**Lemma 3.** *With assumption 2, for heterophilic subgraph $\mathcal{G}^{het}$, when $j \geq N-1$, $\boldsymbol{XWWX} = w_0 + w_1\boldsymbol{L}^{het} + w_2(\boldsymbol{L}^{het})^2 + \cdots + w_j(\boldsymbol{L}^{het})^j = \boldsymbol{Q}_{het}\boldsymbol{A}_{het}\boldsymbol{Q}_{het}^T$, where $\boldsymbol{A}_{het} = diag(\alpha_{\boldsymbol{L}_1} \ldots \alpha_{\boldsymbol{L}_N})$. $\alpha_{\boldsymbol{L}_1} \ldots \alpha_{\boldsymbol{L}_N}$ are $N$ different parameters, if $\lambda_{\boldsymbol{A}_1^{het}} \ldots \lambda_{\boldsymbol{A}_N^{het}}$ are $N$ different frequency amplitudes.*

*Proof.* The proof can be found in Theorem 4 of [Liu et al., 2022], by replacing $\boldsymbol{L}$ as the decomposing matrix. $\qquad\square$

For $\boldsymbol{Z}_l\tilde{\boldsymbol{Z}}_l^T$, using Lemma 2, we have:

$$
\begin{aligned}
\boldsymbol{Z}_l\tilde{\boldsymbol{Z}}_l^T &= \boldsymbol{AXWWX}\tilde{\boldsymbol{A}} \\
&= \boldsymbol{Q}_{\text{hom}}\boldsymbol{\Lambda}_{\text{hom}}\boldsymbol{Q}_{\text{hom}}^T\boldsymbol{Q}_{\text{hom}}\boldsymbol{A}_{\text{hom}}\boldsymbol{Q}_{\text{hom}}^T\boldsymbol{Q}_{\text{hom}}\tilde{\boldsymbol{\Lambda}}_{\text{hom}}\boldsymbol{Q}_{\text{hom}}^T \\
&= \boldsymbol{Q}_{\text{hom}}\boldsymbol{\Lambda}_{\text{hom}}\boldsymbol{A}_{\text{hom}}\tilde{\boldsymbol{\Lambda}}_{\text{hom}}\boldsymbol{Q}_{\text{hom}}^T \\
&= \boldsymbol{Q}_{\text{hom}}
\begin{bmatrix}
\lambda_{\boldsymbol{A}_1^{\text{hom}}}\alpha_{\boldsymbol{A}_1}\lambda_{\tilde{\boldsymbol{A}}_1^{hom}} & 0 & \cdots & 0 \\
0 & \lambda_{\boldsymbol{A}_2^{\text{hom}}}\alpha_{\boldsymbol{A}_2}\lambda_{\tilde{\boldsymbol{A}}_2^{hom}} & \cdots & 0 \\
\vdots & \vdots & \ddots & \vdots \\
0 & 0 & \cdots & \lambda_{\boldsymbol{A}_N^{\text{hom}}}\alpha_{\boldsymbol{A}_N}\lambda_{\tilde{\boldsymbol{A}}_N^{\text{hom}}}
\end{bmatrix}
\boldsymbol{Q}_{\text{hom}}^T \\
&= \sum_{i=1}^{N}\lambda_{\boldsymbol{A}_i^{\text{hom}}}\alpha_{\boldsymbol{A}_i}\lambda_{\tilde{\boldsymbol{A}}_i^{\text{hom}}}q_{\boldsymbol{A}_i}q_{\boldsymbol{A}_i}^T,
\end{aligned}
$$

where $q_{\boldsymbol{A}_i}$ is the $i^{th}$ column of the matrix $\boldsymbol{Q}_{\text{hom}}$.

For $\boldsymbol{Z}_h\tilde{\boldsymbol{Z}}_h^T$, using Lemma 3, we have:

$$
\begin{aligned}
\boldsymbol{Z}_h\tilde{\boldsymbol{Z}}_h^T &= \boldsymbol{LXWWX}\tilde{\boldsymbol{L}} \\
&= \boldsymbol{Q}_{\text{het}}\boldsymbol{\Lambda}_{\text{het}}\boldsymbol{Q}_{\text{het}}^T\boldsymbol{Q}_{\text{het}}\boldsymbol{A}_{\text{het}}\boldsymbol{Q}_{\text{het}}^T\boldsymbol{Q}_{\text{het}}\tilde{\boldsymbol{\Lambda}}_{\text{het}}\boldsymbol{Q}_{\text{het}}^T \\
&= \boldsymbol{Q}_{\text{het}}\boldsymbol{\Lambda}_{\text{het}}\boldsymbol{A}_{\text{het}}\tilde{\boldsymbol{\Lambda}}_{\text{het}}\boldsymbol{Q}_{\text{het}}^T \\
&= \boldsymbol{Q}_{\text{het}}
\begin{bmatrix}
\lambda_{\boldsymbol{L}_1^{\text{het}}}\alpha_{\boldsymbol{L}_1}\lambda_{\tilde{\boldsymbol{L}}_1^{\text{het}}} & 0 & \cdots & 0 \\
0 & \lambda_{\boldsymbol{L}_2^{\text{het}}}\alpha_{\boldsymbol{L}_2}\lambda_{\tilde{\boldsymbol{L}}_2^{\text{het}}} & \cdots & 0 \\
\vdots & \vdots & \ddots & \vdots \\
0 & 0 & \cdots & \lambda_{\boldsymbol{L}_N^{\text{het}}}\alpha_{\boldsymbol{L}_N}\lambda_{\tilde{\boldsymbol{L}}_N^{\text{het}}}
\end{bmatrix}
\boldsymbol{Q}_{\text{het}}^T \\
&= \sum_{i=1}^{N}\lambda_{\boldsymbol{L}_i^{\text{het}}}\alpha_{\boldsymbol{L}_i}\lambda_{\tilde{\boldsymbol{L}}_i^{\text{het}}}q_{\boldsymbol{L}_i}q_{\boldsymbol{L}_i}^T,
\end{aligned}
$$

where $q_{\boldsymbol{L}_i}$ is the $i^{th}$ column of the matrix $\boldsymbol{Q}_{\text{het}}$. Therefore, we have:

$$
\text{tr}(\boldsymbol{Z}_l\tilde{\boldsymbol{Z}}_l^T) = \sum_{i=1}^{N}\lambda_{\boldsymbol{A}_i^{\text{hom}}}\alpha_{\boldsymbol{A}_i}\lambda_{\tilde{\boldsymbol{A}}_i^{\text{hom}}}, \quad sum(\boldsymbol{Z}_l\tilde{\boldsymbol{Z}}_l^T) = \sum_i \lambda_{\boldsymbol{A}_i^{\text{hom}}}\alpha_{\boldsymbol{A}_i}\lambda_{\tilde{\boldsymbol{A}}_i^{\text{hom}}}sum(q_{\boldsymbol{A}_i}q_{\boldsymbol{A}_i}^T)
$$

$$
\text{tr}(\boldsymbol{Z}_h\tilde{\boldsymbol{Z}}_h^T) = \sum_{i=1}^{N}\lambda_{\boldsymbol{L}_i^{\text{het}}}\alpha_{\boldsymbol{L}_i}\lambda_{\tilde{\boldsymbol{L}}_i^{het}}, \quad sum(\boldsymbol{Z}_h\tilde{\boldsymbol{Z}}_h^T) = \sum_i \lambda_{\boldsymbol{L}_i^{\text{het}}}\alpha_{\boldsymbol{L}_i}\lambda_{\tilde{\boldsymbol{L}}_i^{het}}sum(q_{\boldsymbol{L}_i}q_{\boldsymbol{L}_i}^T)
$$

By substituting this into Eq. (21), we have

$$
\begin{aligned}
\mathcal{L}_{HLCL} &\geq -\left(\sum_{i=1}^{N}\left(\lambda_{\boldsymbol{A}_i^{\text{hom}}}\alpha_{\boldsymbol{A}_i}\lambda_{\tilde{\boldsymbol{A}}_i^{\text{hom}}} + \lambda_{\boldsymbol{L}_i^{\text{het}}}\alpha_{\boldsymbol{L}_i}\lambda_{\tilde{\boldsymbol{L}}_i^{\text{het}}}\right) - \frac{1}{N}\sum_{i=1}^{N}\left(\lambda_{\boldsymbol{A}_i^{\text{hom}}}\alpha_{\boldsymbol{A}_i}\lambda_{\tilde{\boldsymbol{A}}_i^{\text{hom}}}\sum(q_{\boldsymbol{A}_i}q_{\boldsymbol{A}_i}^T) + \lambda_{\boldsymbol{L}_i^{\text{het}}}\alpha_{\boldsymbol{L}_i}\lambda_{\tilde{\boldsymbol{L}}_i^{\text{het}}}\sum(q_{\boldsymbol{L}_i}q_{\boldsymbol{L}_i}^T)\right)\right) \\
&= -\left(\sum_{i=1}^{N}\lambda_{\boldsymbol{A}_i^{\text{hom}}}\alpha_{\boldsymbol{A}_i}\lambda_{\tilde{\boldsymbol{A}}_i^{\text{hom}}}\left(1 - \frac{1}{N}\sum(q_{\boldsymbol{A}_i}q_{\boldsymbol{A}_i}^T)\right) + \lambda_{\boldsymbol{A}_i^{\text{het}}}\alpha_{\boldsymbol{L}_i}\lambda_{\tilde{\boldsymbol{A}}_i^{\text{het}}}\left(1 - \frac{1}{N}\sum(q_{\boldsymbol{L}_i}q_{\boldsymbol{L}_i}^T)\right)\right)
\end{aligned}
$$

Since $q_i^T q_i = 1$, $|q_{ij}| < 1$, $\sum(q_i q_i^T) > -N^2$, we have

$$\mathcal{L}_{HLCL} \geq (-1 - N) \sum_{i=1}^{N} \left( \lambda_{\boldsymbol{A}_i^{\text{hom}}} \alpha_{\boldsymbol{A}_i} \lambda_{\tilde{\boldsymbol{A}}_i^{\text{hom}}} + \lambda_{\boldsymbol{A}_i^{\text{het}}} \alpha_{\boldsymbol{L}_i} \lambda_{\tilde{\boldsymbol{A}}_i^{\text{het}}} \right).$$

Since $\lambda_{\boldsymbol{A}_i^{\text{hom}}} \in (-1, 1]$, and $\lambda_{\boldsymbol{L}_i^{\text{het}}} \in [0, 2)$, we have

$$\mathcal{L}_{HLCL} \geq \frac{-1 - N}{2} \sum_{i=1}^{N} \left( \alpha_{\boldsymbol{A}_i} \left( 2 - (\lambda_{\boldsymbol{A}_i^{\text{hom}}} - \lambda_{\tilde{\boldsymbol{A}}_i^{\text{hom}}})^2 \right) + \alpha_{\boldsymbol{L}_i} \left( 4 - (\lambda_{\boldsymbol{L}_i^{\text{het}}} - \lambda_{\tilde{\boldsymbol{L}}_i^{\text{het}}})^2 \right) \right).$$

$\square$

