# OpenReview forum: "Graph Contrastive Learning under Heterophily via Graph Filters"
_auai.org/UAI/2024/Conference — UAI 2024 poster_

### Official Review · Reviewer_tzsP · 2024-03-19

**Q2-1 Originality-Novelty:** 3
**Q2-2 Correctness-Technical Quality:** 3
**Q2-5 Clarity Of Writing:** 2

**Q1 Summary And Contributions:**

The paper proposes a contrastive graph learning approach that copes well with heterophily than existing approaches. Indeed, the authors propose to use both the adjacency and the Laplacian matrices for low-pass and high-pass filtering, thus covering both homophilic and heterophilic subgraphs well. The results seem to support the claims.

**Q2-3 Extent To Which Claims Are Supported By Evidence:**

2: Fair: the main claims are somewhat supported by evidence (but the experimental evaluation may be weak, or does not match entirely with the claims, important baselines may be missing, proofs contain important ideas but lack rigor, algorithmic details are only discussed superficially, references are imprecise, assumptions are not sufficiently motivated or explicated, etc.).

**Q2-4 Reproducibility:**

3: Good: key resources (e.g. proofs, code, data) are available and key details (e.g. proofs, experimental setup) are sufficiently well-described for competent researchers to confidently reproduce the main results.

**Q3 Main Strengths:**

The paper is mostly accessible and the approach is both timely (self-supervised learning has become an important class of learning problems) and valid (the utility of high-frequency graph signals has recently been recognized). The experimental results seem to support the main claims (but see below).

**Q4 Main Weakness:**

Aside from some minor aspects (see below), I have some reservations that prevent me from assigning a better score. I hope that these can be relieved during the rebuttal phase.
- One main drawback is the assumption in Sec. 4.1 that the original features can indicate label information. If this was the case, then the problem can be solved without considering the graph structure. To counter this shortcoming, the authors should show the classification results for a nonlinear classifier (MLP or similar) operating directly on the features $\mathbf{x}$ that is trained with the same train/test/validation split. Further, the assumption should be tested, comparing the homophily ratios of the homophilic and heterophilic subgraphs selected based on the feature similarity.
- In Final Representations, why is only the low-pass filtered representation selected as the final output? This is not clear and seems to be not justified well.
- Minimizing a lower bound is not equivalent to minimizing the bounded quantity; also, in Sec. 4.2 it is not clear if the authors claim that HLCL minimizes the lhs or the rhs of the assertion of Th. 1. This somewhat limits the contribution of the theoretical analysis, but can probably be repaired during the rebuttal.
- It is not fully clear how the hyperparameters of HLCL and the comparison methods were chosen. Further, at least in Table 1 the results of HLCL do not seem to consistently outperform the competing methods. This is not a limitation in itself, but at least should be commented on.

**Q5 Detailed Comments To The Authors:**

- The language of the paper can be improved, as there are some grammatical errors. (E.g., "learning methods has become", "succesful", "To address this, recently HGRL, a graph self-supervised learning method, Chen et al. [2022], proposed...", "GNNs propagations",...)
- In Fig. 1, the notation is not (yet) clear. Also, why do both networks result in the same representation prior to the projection head ($\mathbf{H}_H$)?
- In Sec. 3.1, it is not clear how the entries of eigenvectors connect with clusters of connected nodes. Can you give an intuition or a reference?
- What is the projection head $g$ doing? I did not find it in the main text.
- In Table 1, it is surprising that methods designed for heterophilic graphs do not perform well on these graphs; e.g., the accuracies of MixHop and H2GCN are not much better than those of the remaining methods; and methods like HGRL do not outperform methods like BGRL. Are there any explanations?
- The experiment explained in Table 2 is not fully clear; the caption is confusing, and it is not evident which lines refer to het, hom, and both.
- The insight that the homophily ratio is a good indicator for appropriate values of $k_1$ and $k_2$ is very interesting. Can you provide more details?

### Minor Comments:
- At the end of Sec. 1, a 12% improvement is claimed, which is in conflict with the 14% improvement in the abstract.
- In Sec. 3, why is the edge set defined to be directed, if the graph is undirected?
- The usage of cite, citet, and citep seems to be incorrect in some instances.

**Q9 Complying With Reviewing Instructions:**

Yes

---

> ### Author Rebuttal · Authors · 2024-04-05
>
> We thank the reviewer for acknowledging the effectiveness and validity of our method, backed up by our empirical experiments.
>
> Q1. Features vs labels
>
> We emphasize that, while features can approximately indicate if neighboring nodes are of the same class, they are insufficient for accurate (multi-class) node classification, and graph structure is crucial to take into account. Otherwise one could simply use an MLP classifier on node features. It is important to note that approximately identifying a homophilic and a heterophilic subgraph is a binary classification task, and is significantly easier than multiclass node classification.
>
> First, we show the insufficiency of node features for accurate classification without graph structure. We conducted additional experiments with an MLP classifier on various homophily and heterophily datasets, which showed that MLP yields very poor performances, in particular under heterophily.
>
> ||Cora (6 classes)|Citeseer (7 classes)|Chameleon (5 classes)|Squirrel (5 classes)|
> |---|---|---|---|---|
> |MLP|64.8 ± 1.2|66.5 ± 1.0|37.4 ± 2.1|25.5 ± 0.9|
> |HLCL|84.1 ± 1.0|70.1 ± 0.8|50.9 ± 1.0|42.9 ± 2.6|
>
> Next, we show that it can still approximately give neighborhood information, thus helpful in splitting the subgraph: we provide the homophily ratios of the original graph, the homophilic subgraph, and heterophilic subgraph selected based on the feature similarity, on different datasets. As shown, with feature cosine similarity, HLCL can approximately create homophilic subgraph and heterophilic subgraph from the original graph. However, relying on this approximation can lead to suboptimal performances, as shown in Table 6.3. Instead, HLCL updates subgraphs based on node representations during training and further improves the model performance.
>
> ||Cora(hom)|Citeseer(hom)|Chameleon(het)|Squirrel(het)|
> |---|---|---|---|---|
> |orig graph hom%|0.83|0.71|0.23|0.19|
> |hom subgraph hom%|0.87|0.82|0.74|0.42|
> |het subgraph hom%|0.08|0.05|0.24|0.19|
>
> To summarize, using feature cosine similarity alone is insufficient for accurate node classification, as we confirmed above. Nevertheless, node feature similarity can help identify approximate homophilic and heterophilic subgraphs at the beginning of training. We note that updating these subgraphs based on node representation (instead of feature) similarity is necessary during the training, especially on heterophilic graphs. Otherwise, the performance degrades.
>
> Q2: Low-pass representation for final output
>
> It is important to note that the encoder is trained using contrastive loss on both high-pass filtered heterophilic subgraphs and low-pass filtered homophilic subgraphs. This training ensures that nodes in the same class have similar representations when a low-pass filter is applied, and nodes in different classes have distinct representations with a high-pass filter. During inference, the goal is to identify nodes with similar representations. Hence using low-pass filtered representatives work better in practice, than using the high-pass filtered representations or a combination of both. The following table confirms that across different datasets, using only the low-pass filtered representation yields higher performance compared to using only the high-pass filtered representation or combining both, especially on homophily graphs.
>
> | |Cora|Citeseer|Chameleon|Squirrel|
> |---|---|---|---|---|
> |LP|84.1 ± 1.0|70.1 ± 0.8|50.9 ± 1.0|42.9 ± 2.6|
> |HP|51.9 ±2.9|36.2 ±2.5|35.6 ±3.1|29.8 ±1.6|
> |[LP,HP]|74.2 ± 2|57.6 ± 2|48.7 ± 1|40.8 ± 2|
>
> Q3: Theoretical Analysis
>
> Theorem 1 provides a lower-bound for the HLCL loss, implying that minimizing the HLCL loss equals to minimizing the lower bound. The lower-bound is in the form of a summation of two terms: the first term is the sum of the difference between the low-frequency components of the two low-pass filtered augmented views of the graph, i.e.  $\frac{-1-N}{2}\sum_i \alpha_{A_i} (2- (\lambda_{A_i}^{\text{hom}}-\lambda_{\tilde{A}_i}^{\text{hom}})^2$), and the second term is the sum of the difference between the two high-pass filtered augmented views of the graph, i.e.
>
> $\frac{-1-N}{2}\sum_i \alpha_{L_i}(4-(\lambda_{L_i}^{\text{het}}-\lambda_{\tilde{L}_i}^{\text{het}})^2)$. Minimizing the HLCL loss minimizes the lower bound. In doing so, the entire expression inside the sum (without the negative sign before the sum) is maximized, hence larger $\alpha{A_i}$ will be assigned to the smaller
>
> $(\lambda_{A_i}^{hom}-\lambda_{\tilde{A}_i}^{hom})^2$, hence
>
> $\lambda_{A_i}^{hom} \sim \lambda_{\tilde{A}_i}^{hom}$
>
> Notably, $\lambda_{A_i}^{hom} \sim \lambda_{\tilde{A}_i}^{hom}$ implies that the two contrasted augmentations are invariant at $i^{th}$ frequency. Same reasoning holds for the second term. Therefore, during training with HLCL, the encoder will emphasize the invariance between two contrasted augmentations from the spectrum domain, for both the homophilic and heterophilic subgraphs.

---

### Official Review · Reviewer_8V3g · 2024-03-21

**Q2-1 Originality-Novelty:** 2
**Q2-2 Correctness-Technical Quality:** 3
**Q2-5 Clarity Of Writing:** 3

**Q1 Summary And Contributions:**

The paper proposes an effective graph CL method, namely HLCL, for learning graph representations under heterophily. HLCL uses a low-pass and a high-pass graph filter to aggregate representations of nodes connected in the homophily subgraph and differentiate representations of nodes in the heterophily subgraph and learn node representations by contrasting both the augmented high-pass filtered views and the augmented low-pass filtered node views. HLCL outperforms other comparison methods on node classification tasks on multiple benchmark datasets.

**Q2-3 Extent To Which Claims Are Supported By Evidence:**

2: Fair: the main claims are somewhat supported by evidence (but the experimental evaluation may be weak, or does not match entirely with the claims, important baselines may be missing, proofs contain important ideas but lack rigor, algorithmic details are only discussed superficially, references are imprecise, assumptions are not sufficiently motivated or explicated, etc.).

**Q2-4 Reproducibility:**

2: Fair: key resources (e.g. proofs, code, data) are unavailable but key details (e.g. proof sketches, experimental setup) are sufficiently well-described for an expert to confidently reproduce the main results.

**Q3 Main Strengths:**

1. The experimental results are promising.

**Q4 Main Weakness:**

See Detailed Comments To The Authors

**Q5 Detailed Comments To The Authors:**

1. The motivation behind this paper seems to be unfounded. In my experience, many heterogeneous graph methods have been proposed to solve the problem of aggregating neighbouring nodes of different classes. Compared with many heterogeneous graph methods, the accuracy of node classification in the author's method is also relatively low. The author needs to clarify the advantages of the method proposed in this paper compared to other heterogeneous graph methods, and not only compare it with traditional Homogeneous graph methods.

2. The author did not explain the specific meaning of some mathematical symbols in the formula. For example, did sigma in Eqs. 4 and 5 not specify its specific meaning?

3. Fig. 1 needs to be redrawn, which makes it difficult to obtain any useful information from Fig. 1. Many details in the paper are not reflected in Fig. 1. For example, how augmented views are obtained, and how anchor embeddings, positive samples, and negative samples are selected in nodes are not reflected in the graph.

4. The author did not compare some of the latest methods in the experiment, and some methods from 2023 should be compared. In addition, the author's experimental comparison setting is unreasonable, and the author should compare some methods for heterogeneous graphs [1], [2], [3], [4].
[1] Zhu J, Rossi R A, Rao A, et al. Graph neural networks with heterophily[C]//Proceedings of the AAAI conference on artificial intelligence. 2021, 35(12): 11168-11176.
[2] Luan S, Hua C, Lu Q, et al. Revisiting heterophily for graph neural networks[J]. Advances in neural information processing systems, 2022, 35: 1362-1375.
[3] Luan S, Hua C, Xu M, et al. When Do Graph Neural Networks Help with Node Classification? Investigating the Homophily Principle on Node Distinguishability[J]. Advances in Neural Information Processing Systems, 2024, 36.
[4] Li X, Zhu R, Cheng Y, et al. Finding global homophily in graph neural networks when meeting heterophily[C]//International Conference on Machine Learning. PMLR, 2022: 13242-13256.

5. The author only compared 7 node classification datasets in the experiment, which is very insufficient. The Pubmed, Texas, Wisconsin, Cornell, pokec, arxiv-year, snap agents, and genius datasets should also be compared.

6. In Formula 8, the author calculated four similarity losses, but no experiments were conducted to verify their effectiveness in the ablation experiment.

**Q9 Complying With Reviewing Instructions:**

Yes

---

> ### Author Rebuttal · Authors · 2024-04-05
>
> We thank the reviewer for recognizing the strong performance of our method.
>
> > Regarding reproducibility, as mentioned by reviewer vin6, our code has been provided in the supplement of the original submission and our results are reproducible.
>
> > motivation of our paper, performance and baselines
>
> There might be a confusion about our method. Our method is a self-supervised method that is capable of learning node representations without any label. In doing so, after training, one can train a linear classifier with a very small number of labels to label all the nodes. As discussed in the introduction (last sentence of first paragraph: “obtaining high-quality labels can be costly in many domains, spurring interest in self-supervised learning on graphs to learn representations without supervision.”) This setting alleviates the need for obtaining labels that are expensive in many scenarios, and hence has gained a lot of attention recently [5-8]. Without labels, how to aggregate information in different neighborhoods under heterophily is indeed very challenging. Our method targets this setting, for heterophilic graphs.
>
> Nevertheless, to show the effectiveness of our method, we have benchmarked our approach against recent supervised methods like H2GCN and MixHop, which are designed specifically for heterophilic graphs. Table 1 shows that HLCL outperforms these supervised methods in most cases. Note that, compared to self-supervised methods, supervised approaches have access to label information during training, allowing them to aggregate information more accurately based on node labels. On the contrary, HLCL learns the node representation without any label. We note that for a fair comparison, and consistent with prior graph CL works [5,7], we use a 10/10/80 split, i.e. we use 10% labels when training supervised methods. For SSL methods, we use no label during training. After training with SSL methods, to evaluate the performance of the methods, 10% of the labels are used to train a linear classifier on the learned node representation (by HLCL and other SSL methods, namely DGI, BGRL, GRACE, SP-GCL, HGRL). Supervised methods heavily rely on labels, hence their performance is lower than using a 48/32/20 split.
>
> Here, we include additional baselines from the papers mentioned by the reviewer, published between 2021 and 2024 [1-4]. While HLCL has lower performances on Cora and CiteSeer which are homophilic graphs, it outperforms all existing supervised methods (CPGNN, GloGNN, H2GCN, MixHop) under heterophily. Besides, HLCL outperforms supervised H2GCN and MixHop on all datasets (except Penn94), under both homophily and heterophily.
>
> |Model|Cora(hom)|Citeseer(hom)|Chameleon(het)|Squirrel(het)|
> |---|---|---|---|---|
> |CPGNN [1]|83.6±1.3|72.1±0.5|33.0±3.2|30.0±2.1|
> |GloGNN [4]|88.3±1.1|77.2±1.8|25.9±3.6|35.1±1.2|
> |HLCL|84.1±1.0|70.1±0.8|50.9±1.0|42.9±2.6|
>
> > Sigma function in Eq 4.5
>
> The sigma function is the activation function, as we mentioned 3 lines after Eq. (5).
>
> > Detailed/clearer figure
>
> Thanks for the feedback, we will redraw the figure to reflect this detailed information.
>
> > Similarity terms in Eq. 8
>
> First, we want to clarify that the loss term in Eq. (8) consists of two symmetric contrastive losses: one for high-pass filtered representations $(l(z_h^i, \tilde{z}_h^i) + l(\tilde{z}_h^i, z_h^i))$ and another for low-pass filtered representations $(l(z_l^i, \tilde{z}_l^i) + l(\tilde{z}_l^i, z_l^i) )$, where $z$ and $\tilde{z}$ are two augmented views of the same node. Each contrastive loss is applied to a different subgraph.
>
> For contrastive learning, we first generate a pair of augmented node views for every node, then for the heterophilic subgraph, we first consider the first augmented view, $z$, as the anchor to get $l(z_h^i, \tilde{z}_h^i)$, and then consider the second augmented view, $\tilde{z}$ as the anchor to get $l(\tilde{z}_h^i, z_h^i)$. In a similar manner, we get two terms for the homophilic subgraph. This symmetric contrastive loss is a standard practice in both graph contrastive learning (Eq 2 of [5]) and contrastive learning on images (Eq 1 of [8]). We note that in Eq. (8), the loss should be normalized by ¼ (not ½), considering there are four loss terms. We will correct this in our revision.
>
> For scenarios where contrastive learning is applied exclusively to either low-pass or high-pass filtered representations, we conducted new ablation studies. The results, as presented below, demonstrate that utilizing both representations from both frequency terms is crucial for HLCL's success.
> |Model|Cora|Citeseer|Chameleon|Squirrel|
> |---|---|---|---|---|
> |HLCL|84.1±1.0|70.1±0.8|50.9±1.0|42.9±2.6|
> |HP|32.5±2.0|26.6±3.1|33.1±3.9|33.1±2.1|
> |LP|83.7±0.7|71.4±1.0|35.4±3.6|36.2±2.8|

---

### Official Review · Reviewer_vin6 · 2024-03-22

**Q2-1 Originality-Novelty:** 3
**Q2-2 Correctness-Technical Quality:** 3
**Q2-5 Clarity Of Writing:** 3

**Q1 Summary And Contributions:**

The paper introduces a new method for contrastive learning on graphs. The setting is unsupervised learning, where we seek to extract relevant node encodings without knowing the vertex labels. To overcome the challenge of heterophily (adjacent nodes with different labels), a new algorithm called HLCL is proposed, which leverages both high-frequency and low-frequency graph signals. By extracting a homophily subgraph and a heterophily subgraph, it is able to contrast two kinds of augmented node representations, thus capturing more of the relationships between nodes. Experiments on various datasets confirm that HLCL achieves high performance when confronted to either homophily or heterophily.

**Q2-3 Extent To Which Claims Are Supported By Evidence:**

4: Excellent: all claims are supported by very convincing evidence (in the form of comprehensive experimental evaluation, rigorous mathematical proofs, detailed (pseudo-)code, precise references, well-motivated and realistic assumptions) and the authors deliver what they promise.

**Q2-4 Reproducibility:**

4: Excellent: key resources (e.g. proofs, code, data) are available and key details (e.g. proof sketches, experimental setup) are comprehensively described for competent researchers to confidently and easily reproduce the main results.

**Q3 Main Strengths:**

**Originality-Novelty**

The authors suggest a new method which clearly advances the state of the art by taking the "best of both worlds": high- and low-frequency information thanks to a pair of well-chosen subgraphs. HLCL improves accuracy in heterophilic graphs without degrading it in homophilic graphs.

I am not at all familiar with contrastive learning, so perhaps my positive view of the paper's originality is biased.

**Correctness-Technical Quality**

The paper displays solid understanding of graph signal processing and spectral analysis.

**Extent To Which Claims Are Supported By Evidence**

This is the main strength of the article: thorough, varied and transparent experiments on 7 real datasets (2 homophilic and 7 heterophilic). The performance is improved compared to the state of the art in most cases, but I am particularly pleased by the ablation study, which shows contribution from all components of HLCL. Appendix 8.3 is also very interesting to show that "just adding a high pass filter" would not be enough.

**Reproducibility**

Code is provided with the paper, it is only missing a README and statement of dependencies to be fully reproducible.

**Clarity Of Writing**

The style was clear, not overly verbose, and the figures helpful.

**Q4 Main Weakness:**

**Correctness-Technical Quality**

My only concern is with the theoretical analysis of Section 4.2. I do not understand it (what are the "adaptive weights" $\alpha$ in theorem 1?), and the statement "HLCL minimizes the lower bound of HLCL loss, which implies that minimizing the loss is equivalent to minimizing the lower bound" seems overly bold.
In the present state I am not sure this section is helpful, but it wouldn't be necessary for the paper to be worthy of acceptation anyway.

**Clarity Of Writing**

Being unfamiliar with contrastive learning, I initially had a lot trouble understanding what the paper was even about.
Some terminology like "contrasting", "augmented", "views", "anchor" appears early on without explanations.
I could only dive into it after skimming onr or two external resources, and in particular the blog post https://lilianweng.github.io/posts/2021-05-31-contrastive/#common-setup helped clarify the setting.
I think contrastive learning deserves a more detailed introduction to help readers ease into the domain.

**Q5 Detailed Comments To The Authors:**

Remarks:

- Page 1: "Indeed, for learning rich representations in graphs with heterophily, contrasting augmented views of the same node is not enough, but it is crucial to differentiate representation of node with different labels" => Sounds like it would be useful in homophilic graphs too?
- Page 2, figure 1: I suggest adding the two graph views to the diagram. At the moment it looks like their paths split within the encoder, but in reality it happens one step before.
- Page 2, figure 1: What does the projection head do? I don't think it is mentioned later.
- Page 3: How do you define "similar" in the homophily ratio? Same label?
- Page 3: Why is it so important that the high-freq and low-freq signals are processed by the _same encoder_?
- Page 3: "we rely on the important observation that for graphs with different homophily ratios, the original features can approximately indicate the label information [Jin et al., 2021]." => Is that commonly accepted?
- Page 5: "To generate the high-pass filtered node views, HLCL leverages the normalized Laplacian matrix of the augmented heterophily subgraph and normalized adjacency matrix of the augmented homophily subgraph" => Why not take those of the initial graph?
- Page 5: "For every node i, we treat one of the augmented projected representation..." => What does projected mean here? The word is present for $z$ but not for $\tilde{z}$ so I thought it was an important distinction
- Page 5: "The second term in the denominator represent the inter-view negative pairs, which are between the anchored view of node i and the views of all other nodes from the other view." => Unclear
- Page 6: Could you please clarify the theoretical analyis? At the moment I cannot wrap my head around it
- Page 6, figure 3: Which Jacobian matrices are we talking about?
- Page 7, table 1: What is the performance metric?
- Page 7: "In addition, we also include popular general graph supervised learning methods like GCN" => Their performance is sometimes superior, especially on Penn94. Are they trained in a supervised fashion?
- Page 8: "The performance of the Cora dataset varies by 30% with different k values, while the performance of the Chameleon dataset varies by 7%. Based on the results, the homophily ratio of the graph is a good indicator of the appropriate k1 (and k2) values" => based on 2 samples?
- Page 12: The related work section in the appendix has significant overlap with the one in the main text
- Page 14: What are the $w_i$ in the assumptions?
- Page 14: "Following [Liu et al., 2022], we assume τ = 1, sim is dot product similarity, and the filtering process is a single message passing without non-linear activation." => If this is your hypothesis, it should be mentioned in the theorem body

Minor typos and form:

- In many places you use "homophily" and "heterophily" as adjectives, even though they are nouns. The proper adjectives are "homophilic / heterophilic"

**Q9 Complying With Reviewing Instructions:**

Yes

---

> ### Author Rebuttal · Authors · 2024-04-05
>
> We thank the reviewer for the careful review of our work, and acknowledging the novelty of our method, the technical soundness of our analysis, the thoroughness of our experiments that substantiate our approach, the clarity of our writing, and our commitment to reproducibility.
>
> >  theoretical analysis of Section 4.2.
>
> Theorem 1 provides a lower-bound for the HLCL loss, implying that minimizing the HLCL loss equals to minimizing the lower bound. The lower-bound is in the form of a summation of two terms: the first term is the sum of the difference between the low-frequency components of the two low-pass filtered augmented views of the graph, i.e.  $\frac{-1-N}{2}\sum_i \alpha_{A_i} (2- (\lambda_{A_i}^{\text{hom}}-\lambda_{\tilde{A}_i}^{\text{hom}})^2$), and the second term is the sum of the difference between the two high-pass filtered augmented views of the graph, i.e.
>
> $\frac{-1-N}{2}\sum_i \alpha_{L_i}(4-(\lambda_{L_i}^{\text{het}}-\lambda_{\tilde{L}_i}^{\text{het}})^2)$. Minimizing the HLCL loss minimizes the lower bound. In doing so, the entire expression inside the sum (without the negative sign before the sum) is maximized, hence larger $\alpha{A_i}$ will be assigned to the smaller
>
> $(\lambda_{A_i}^{hom}-\lambda_{\tilde{A}_i}^{hom})^2$, hence
>
> $\lambda_{A_i}^{hom} \sim \lambda_{\tilde{A}_i}^{hom}$
>
> Notably, $\lambda_{A_i}^{hom} \sim \lambda_{\tilde{A}_i}^{hom}$ implies that the two contrasted augmentations are invariant at $i^{th}$ frequency. Same reasoning holds for the second term. Therefore, during training with HLCL, the encoder will emphasize the invariance between two contrasted augmentations from the spectrum domain, for both the homophilic and heterophilic subgraphs.
>
> > CL explanation
>
> Thank you so much for your suggestions. We will include a short overview of foundational contrastive learning in our preliminary section.
>
> Questions
> (Page 1) Indeed this is useful for homophilic graphs as well. As shown in table 3, with ideal subgraphs, HLCL can improve the performance on homophilic graphs as well. However, this is even more crucial under heterophily. In homophilic graphs, since most nodes in a neighborhood share the same label, simple aggregation without differentiation can still yield strong performance. While under heterophily, nodes in a neighborhood have different labels, hence without differentiating nodes in different classes, contrasting augmented views of the same node does not learn meaningful representations.
>
> (Page 2, figure 1: adding the two graph views) We thank the reviewer for the suggestion on figure 1. We will modify the figure as suggested in our revision.
>
> (Page 2, figure 1: projection head)  A projection head is a two-layer MLP with non-linear activations that is attached to the model during training to project the representations into a space where contrastive loss is applied. After training, the projection head is discarded and a linear model is trained on the pre-projection representations. Projection head is a standard technique used in almost all contrastive learning methods on images [4-5] and graphs [1,8]. We will add the discussion to our revised version.
>
> (Page 3: similar" in homophily ratio) Yes, that refers to the same labels, we’ll modify it to “same label”.
>
> (Page 3: Why same encoder?)  As our theorem indicates, maintaining the invariance of both high-frequency and low-frequency components in the graph representation is crucial. Therefore, it is important to use a shared encoder to ensure that it is optimized by both high-pass and low-pass losses. Empirically, we have also observed that a shared encoder has better performance. Additionally, we want to train a single model that can process both homophilic and heterophilic graphs, rather than multiple models.
>
> (Page 3: features indicate labels) [3] showed that reconstructing node neighborhoods based on k-NN (connecting each node with its k most similar nodes), followed by applying GNNs, can yield surprisingly good results for node classification, particularly on graphs with a mix of heterophilic and homophilic neighborhoods. This suggests that feature similarity can be a useful indicator of label information. Similar methods have also been used in [2], [6], and [7].
>
> (Page 5: leveraging L/A of subgraphs) Laplacian and adjacency matrices indicate how the information in a neighborhood should be aggregated by the GCN encoder. Indeed, it is important to use the corresponding matrices in the subgraphs to (1) pull together representations of nodes with similar neighbors (via low-pass filter on homophilic subgraphs) and (2) push away representations of nodes with different labels (via high-pass filter on heterophilic subgraph). Otherwise representations of all the nodes within a neighborhood will be pulled together and pushed apart at the same time.

---

### Official Review · Reviewer_o8bb · 2024-03-23

**Q2-1 Originality-Novelty:** 2
**Q2-2 Correctness-Technical Quality:** 3
**Q2-5 Clarity Of Writing:** 2

**Q1 Summary And Contributions:**

Graph contrastive learning perform poorly on graphs with heterophily. This work addresses this problem by proposing an effective graph CL method, namely HLCL, for learning graph representations under heterophily.

**Q2-3 Extent To Which Claims Are Supported By Evidence:**

3: Good: the main claims are supported by convincing evidence (in the form of adequate experimental evaluation, proofs, (pseudo-)code, references, assumptions).

**Q2-4 Reproducibility:**

2: Fair: key resources (e.g. proofs, code, data) are unavailable but key details (e.g. proof sketches, experimental setup) are sufficiently well-described for an expert to confidently reproduce the main results.

**Q3 Main Strengths:**

1.	The overall presentation of this paper is good. I can follow the ideas, methods and details of this paper without much efforts.
2.	This work demonstrates superior performance, particularly on some datasets with heterophily.

**Q4 Main Weakness:**

1.	The paper doesn’t clearly show how to sample a homophily and a heterophily subgraph. For example, the heterophily subgraph may be disconnected graph when the original graph has high homophily, vice versa. This point is very important to me.
2.	This work aims to improve the performance on graph with heterophily. However, the performance on Actor is not good, which is the dataset with highest heterophily. Why?

**Q5 Detailed Comments To The Authors:**

Some small mistakes: 1. In Table 4, cora and chameleon should be Cora and Chameleon. 2. The structure of Table 3 is a little strange. 3. In Table 1, the line between Homophily and Heterophily is not aligned. 4. There is a strange dot above Section 5 EXPERIMENTS.

**Q9 Complying With Reviewing Instructions:**

Yes

---

> ### Author Rebuttal · Authors · 2024-04-05
>
> We thank the reviewer for recognizing the clarity of our paper as well as the strong performance of our method.
>
> > Regarding reproducibility, as mentioned by reviewer vin6, our code has been provided in the supplement of the original submission and our results are reproducible.
>
> > how to sample a homophily and a heterophilic subgraph.
>
> To sample homophily and heterophilic subgraphs, we implemented a two-step process: First, for each node, we select the top ceil(k1 fraction of edges) with highest cosine similarity to be in the homophilic subgraph. Then, we select the top ceil(k2 fraction of edges) with lowest cosine similarity for the heterophilic subgraph. Since HLCL uses two contrastive losses, one is applied to the homophilic subgraph and the other is applied to the heterophilic subgraph, only one subgraph needs to be mostly connected to achieve a satisfactory performance (note that the two subgraphs are not contrasted with each other). For example, in a homophilic graph like Cora, the heterophilic subgraph is small (since k2 is small), but the homophilic subgraph contains all the nodes in the original largest connected component of the graph (since k1 is large). Hence other contrastive loss applied to the homophilic subgraph mostly determines the final results and the second contrastive loss minimally affects the performance.
>
> The table below shows the fraction of nodes in the largest connected component of the original graph that are in the largest connected component of each subgraph. We see that under homophily (Cora, Citeseer), all the nodes are in the homophilic subgraphs and the heterophilic subgraph is small and minimally affects the performance. Under extreme heterophily (Actor) almost all the nodes are in the heterophilic subgraph and the homophilic subgraph is small and minimally affects the performance. For other graphs, depending on the tuned value of k1, the size of the largest connected component of the two subgraphs changes.
>
> |Data|Cora (.83)|Citeseer (.71)|Chameleon (.23)|Squirrel (.19)|Actor (.09)|
> |-|-|-|-|-|-|
> |homophilic|1|1|25.6%|94.4%|4.7%|
> |heterophilic|8.5%|7.7%|98.2%|95%|99.8%|
>
> >Performance on Actor is not good. Why?
>
> We note that HLCL surpasses all the baselines on the Actor dataset, except HGRL by only 1.4% and GRACE by 0.4%. Please also note the large std of HGRL and GRACE that are 0.9% and 1.1% respectively, compared to the std of HLCL that is 0.2%. We see that HLCL is within 1 standard deviation of GRACE and 2 standard deviations of HGRL, and outperforms these methods on other datasets. We also note that our reported performance for GRACE is higher than what is reported in [2].
>
> This is because, as noted in Table 5 of [1], for the Actor dataset, nodes from different classes have similar connectivity patterns to other classes. That is, nodes in the Actor dataset are connected to nodes from their own class to the same extent that they are connected to nodes from other classes. In this case, the graph structure is not useful to classify the nodes (and may hurt the performance), and only relying on node features achieve a better performance. As shown in [1] and [2], models like MLP which do not use any graph structure can outperform GNN methods like GCN and even H2GCN on the Actor dataset. HGRL uses MLP as its encoder and does not leverage the graph structure, hence it can slightly outperform HLCL on Actor, but is outperformed by HLCL on other datasets.
>
> > Small mistakes
>
> Thank you for pointing out the mistakes.  We will fix these in the revised version.
>
>
> [1]"Is homophily a necessity for graph neural networks?" [arXiv 2021]
> [2] "Towards self-supervised learning on graphs with heterophily." [CIKM 2022]

---

### Official Review · Reviewer_RNdM · 2024-03-25

**Q2-1 Originality-Novelty:** 2
**Q2-2 Correctness-Technical Quality:** 3
**Q2-5 Clarity Of Writing:** 3

**Q1 Summary And Contributions:**

This work proposes a graph contrastive learning framework, namely HLCL, to enhance performance under heterophily. The authors started with the motivation that the existing contrastive learning methods perform poorly on graphs with heterophily. HLCL first measures the distance between node features to split a homophily subgraph and a heterophily subgraph.  Next, graph filters are used to aggregate and differentiate the node representations on different sub-graphs, respectively.

**Q2-3 Extent To Which Claims Are Supported By Evidence:**

3: Good: the main claims are supported by convincing evidence (in the form of adequate experimental evaluation, proofs, (pseudo-)code, references, assumptions).

**Q2-4 Reproducibility:**

2: Fair: key resources (e.g. proofs, code, data) are unavailable but key details (e.g. proof sketches, experimental setup) are sufficiently well-described for an expert to confidently reproduce the main results.

**Q3 Main Strengths:**

1. The experimental results can relatively support the authors' claims. The proposed HLCL achieves performance improvement under heterophily and maintains comparable performance against SOTA under homophily.
2. The proposed HLCL does not rely on supervised information and improves the learning of node representations through self-supervision.

**Q4 Main Weakness:**

1. The work should clarify their contribution, comparing it with the existing self-supervised method.
2. More discussion of the limitations of HLCL. The present experimental setup demonstrates the effectiveness of HLCL, but whether the setup is reasonable and under what heterogeneous graph data HLCL performance degrades.

**Q5 Detailed Comments To The Authors:**

It is recommended to optimize the motivation in the Introduction section.

**Q9 Complying With Reviewing Instructions:**

Yes

---

> ### Author Rebuttal · Authors · 2024-04-05
>
> We thank the reviewer for their positive evaluation of our work and for acknowledging the thoroughness of our experiments and the strong performance of our method, which achieves state-of-the-art (SOTA) performance under heterophily and maintains comparable results against SOTA under homophily.
>
> Regarding reproducibility, as mentioned by Reviewer vin6, our code is available in the supplement of the original submission, ensuring that our results are reproducible.
>
> Our contribution compared to other existing self-supervised learning (SSL) methods is threefold:
> 1. **HLCL as a Pioneering Method:** HLCL is the first SSL method to utilize graph filters, combining high-pass and low-pass filtered representations using a contrastive loss. This approach enables HLCL to learn rich representations under heterophily.
> 2. **Unique Aggregation Strategy:** HLCL identifies two subgraphs with high and low homophily ratios for effective information aggregation. This strategy is unique to HLCL among existing SSL methods.
> 3. **Theoretical Groundwork:** We provide the first theoretical framework for graph contrastive learning (CL) under heterophily, which we hope will pave the way for more comprehensive theoretical analyses of SSL from graphs.
>
> > **Limitations of HLCL**
>
> **Setup**: Our experiments employ commonly used graph baselines and adhere to widely accepted experimental settings, as referenced in [1-4], ensuring that our setup is reasonable.
>
> **Performance**: The performance of learning from heterophily graphs heavily depends on how information is aggregated in different neighborhoods. Label availability is crucial for guiding this aggregation. In the absence of labels, a significant challenge for any graph SSL method, including HLCL, arises when nodes with similar labels cannot be approximately identified.
>
> For instance, in the Penn94 dataset, all SSL methods in Table 1, including HLCL, underperform compared to supervised methods. This underperformance is due to the lack of correlation between node feature similarity and class label similarity in this dataset. In contrast, in the Cora dataset, nodes belonging to the same class exhibit an average of 31% higher feature cosine similarity than nodes from different classes, while in Chameleon, this difference is 11%. However, in Penn94, the difference is only 1.4% on average, indicating a high similarity in node features across different classes. Consequently, SSL methods, including HLCL, face challenges in learning high-quality node representations. Despite this, HLCL outperforms other SSL baselines on Penn94, as shown in Table 1. We will include this discussion in our revised version.
>
> References:
>
> [1] "Deep graph contrastive representation learning." [preprint 2020]
>
> [2] Chen, J., et al. "Towards self-supervised learning on graphs with heterophily." [CIKM 2022]
>
> [3] Wang, H., et al. "Can Single-Pass Contrastive Learning Work for Both Homophilic and Heterophilic Graph?" [preprint 2022]
>
> [4]Zhu, J., et al. "Graph neural networks with heterophily." [AAAI 2021]

---

### Meta-Review · Area_Chair_cMZe · 2024-04-16

Graph neural networks are known to perform poorly on graphs with heterophily. The paper  proposes a graph contrastive learning framework, called HLCL, as a remedy to this problem. HLCL outperforms in general other benchmarks. The reviewers find that the paper contains an interesting, novel contribution following an approach that makes sense, and the rebuttal cleared most of their initial objections.